# TRIM2 inhibits apoptosis by ubiquitinating BNIP3 to protect the intestine against ischemia-reperfusion injury in mice
Jinping Nie[1,4], Chao Mei[1,2,4], Aiping Wei[1], Yingjie Wang[1], Chenlu Fan[1], Yingjie Huang[1], Ming Jiang[1], Han Che[1], Tao Chen [1], Juan Tian[1], Yong Li [1] ✉, Xuan Huang [3] ✉ & Xuekang Zhang [1] ✉

Intestinal damage following interrupted blood flow and its return (intestinal ischemia/reperfusion injury) is a serious medical problem occurring in various clinical situations. While the death of intestinal epithelial cells is a key factor, the precise reasons behind this cell death are not fully known. In this study, we identified significant downregulation of an E3 ubiquitin ligase TRIM2 in mouse models of this injury and in cells mimicking the condition. Genetic deletion of TRIM2 promotes intestinal apoptosis and worsens injury severity in studies using only male mice. We discovered that TRIM2 directly interacts with the pro-apoptotic protein Bcl2-interacting protein 3 (BNIP3) and mediates K48-linked polyubiquitination of BNIP3 at lysine 130 (K130), leading to its proteasomal degradation. Mutation of BNIP3 at K130 to arginine (K130R) abolished TRIM2-mediated ubiquitination, increased BNIP3 stability, and led to increased cell death after oxygen deprivation and restoration (hypoxia/ reoxygenation). Increasing BNIP3 levels counteract the protective effect of boosting TRIM2 in intestinal epithelial cells, while lowering BNIP3 mimics the protection seen with more TRIM2. Therefore, TRIM2 protects against intestinal injury by inhibiting apoptosis through the ubiquitination and degradation of BNIP3. Targeting this TRIM2-BNIP3 axis offers possibilities for developing future treatments for intestinal ischemia/reperfusion injury.

Intestinal ischemia/reperfusion (II/R) injury is a critical acute condition characterized by disruption of the intestinal barrier and bacterial translocation, which can precipitate systemic inflammatory response syndrome (SIRS) and multiple organ dysfunction syndrome (MODS)[1]. This pathology frequently arises in clinical scenarios such as intestinal obstruction, transplantation, torsion, and severe trauma surgery, all of which are driven by prolonged ischemia[2]. The pathogenesis of II/R injury involves two distinct phases: ischemic insult, which directly impairs cellular integrity, and reperfusion injury, which exacerbates damage through oxidative stress and the inflammatory cascades[3]. These processes collectively degrade cellular lipids, proteins, and DNA, destabilize intestinal epithelial homeostasis, and trigger apoptosis[4]. Despite its clinical significance, II/R injury lacks safe and effective therapies, with mortality rates in the critical care setting ranging from 67% to 80%[5]. Thus, elucidating its pathophysiology and identifying therapeutic targets are essential.

The mechanisms underlying II/R injury are multifaceted, encompassing apoptosis, inflammation, oxidative stress, necroptosis, autophagy, and ferroptosis[4]. Among these, apoptosis, a programmed cell death, emerges as a central pathological contributor to II/R injury[4]. This tightly regulated process, governed by genetic mechanisms, occurs in both physiological and pathological contexts[6] and is orchestrated by intricate signaling pathways involving specific protein families or genes[7]. Targeting these key regulators represents a promising strategy for mitigating II/R-induced damage.

Ubiquitination, a ubiquitous post-translational modification, plays a pivotal role in disease pathogenesis by modulating protein stability and function[8]. E3 ubiquitin ligases confer specificity to this process by conjugating ubiquitin to substrate proteins, marking them for degradation or functional alteration[9]. Recent evidence highlights ubiquitination's role in II/ R injury, particularly as a regulator of apoptosis[10,11]. The tripartite motif-containing (TRIM) protein family, the largest group of E3 ubiquitin ligases,

[1]Department of Anesthesiology, The First Affiliated Hospital, Jiangxi Medical College, Nanchang University, Nanchang, 330006, China. [2]Department of Surgery and Anesthesia, Ganjiang New Area Hospital of The First Affiliated Hospital of Nanchang University, Ganjiang New Area People's Hospital, Nanchang, 330029, China. [3]The National Engineering Research Center for Bioengineering Drugs and the Technologies, Jiangxi Provincial Key Laboratory of Bioengineering Drugs, Institute of Translational Medicine, Jiangxi Medical College, Nanchang University, Nanchang, 330031, China. [4]These authors contributed equally: Jinping Nie, Chao Mei. ✉e-mail: liyong@ncu.edu.cn; huangxuan@ncu.edu.cn; ndyfy00768@ncu.edu.cn

participates in diverse biological processes, including cell proliferation, apoptosis, inflammation, and innate immunity[12–14]. Notably, TRIM2, a member of this family, has been associated with neuroprotection in ischemic tolerance via interaction with Bim[15]. However, its contributions to II/R injury have remained uncharted.

In this study, we observed a marked downregulation of TRIM2 expression in both in vivo and in vitro II/R models. Our results reveal that TRIM2 exerts a protective effect against II/R injury by suppressing apoptosis in intestinal epithelial cells. Mechanistically, TRIM2 negatively regulates Bcl2-interacting protein 3 (BNIP3) by catalyzing K48-linked poly-ubiquitination at lysine 130 (K130), thereby promoting its degradation. These findings identify the TRIM2-BNIP3 axis as a therapeutic target for II/R injury, providing fresh insights into its molecular basis and potential treatment avenues.

## Results

### TRIM2 expression is downregulated in both in vivo and in vitro models of II/R

Emerging evidence highlights the critical involvement of TRIM family proteins in acute inflammatory and apoptotic pathways[16,17], yet their role in II/R injury remains poorly characterized. To identify potential candidates, we performed qRT-PCR screening of TRIM family members in a murine II/R model. TRIM2 emerged as the most significantly dysregulated gene, showing marked downregulation (Fig. 1A). This suppression was further validated in rat intestinal epithelial cells (IEC-6) subjected to H/R (Fig. 1B). Consistent with transcriptional changes, Western blot analysis revealed a progressive decline in TRIM2 protein expression in IEC-6 cells, beginning at 1 h post-hypoxia and persisting through 24 h (Fig. 1C). Similar results were observed in human Caco-2 cells (Figs. 1D and 1E). Histological examination using the H&E staining method revealed that the intestinal villi of mice exhibited substantial damage following II/R injury, including detachment of the intestinal villi and lamina propria, as well as bulging of the intestinal mucosal epithelium (Figs. 1F and 1G). IHC staining results corroborated these findings, showing a marked reduction in TRIM2 expression in injured tissues (Fig. 1F and 1H). Western blot analysis confirmed the significant downregulation of TRIM2 in the intestines of II/R mice, accompanied by alterations in the expression of apoptotic proteins Bax and Bcl2 (Fig. 1I, see also Supplementary Fig. 1A). It is noteworthy that the mRNA level of TRIM2 was found to be significantly reduced as well (Supplementary Fig. 1B). Additionally, immunofluorescence analysis of TRIM2 in Caco-2 and IEC-6 cells revealed that TRIM2 expression was predominantly cytoplasmic and significantly reduced following H/R treatment (Fig. 1J). Collectively, these data identify TRIM2 as a dynamically regulated protein in II/R injury, with its suppression correlating strongly with apoptotic activation and tissue damage.

### The Knockout of TRIM2 exacerbates II/R-induced apoptosis in the intestine

To delineate the functional role of TRIM2 in II/R injury, we employed the CRISPR/Cas9 system to generate *Trim2*[-/-] mice (Supplementary Fig. 2). H&E staining revealed that II/R caused significant intestinal mucosal damage, including disintegration of intestinal villi, detachment of epithelial cells, and denudation of villus tips (Fig. 2A). It is noteworthy that TRIM2 deficiency markedly exacerbated II/R-induced intestinal mucosal injury in comparison to WT mice (Fig. 2A, see also Supplementary Fig. 3A). TUNEL staining revealed a significant increase in the proportion of apoptotic cells in *Trim2*[-/-] mice compared to WT mice, indicating that TRIM2 deficiency promotes apoptosis during II/R injury (Fig. 2B, see also Supplementary Fig. 3B). To gain further insight into the role of TRIM2 in apoptosis regulation, IHC and Western blot analyses were conducted to evaluate the expression of pro-apoptotic proteins (Bax, Bad, and cleaved Casp3) and the anti-apoptotic protein Bcl2. The II/R treatment resulted in an increase in the expression of Bax, Bad, and cleaved Casp3, accompanied by a decrease in the expression of Bcl2 (Fig. 2C, D, see also Supplementary Fig. 3D-E). These changes were more pronounced in *Trim2*[-/-] mice (Fig. 2C, D, see also

Supplementary Fig. 3D, E). Similarly, the mRNA expression levels of Bax, Bcl2, and Bad exhibited the same patterns as their protein counterparts (Supplementary Fig. 3C).

Given that mitochondrial dysfunction is a hallmark of apoptosis[18], transmission electron microscopy (TEM) was employed to examine mitochondrial morphology in intestinal epithelial cells following II/R injury. In the WT sham and *Trim2*[-/-] sham groups, mitochondria exhibited normal, intact structures (Fig. 2E). However, following II/R treatment, mitochondria displayed signs of damage, appearing swollen with reduced and disorganized cristae (Fig. 2E). Mitochondrial damage was more severe in *Trim2*[-/-] mice, with extensive swelling and partial fragmentation of the organelles (Fig. 2E). Mitochondrial autophagy plays a pivotal role in the pathogenesis of II/R injury[4]. Our findings indicate that II/R effectively induces an increase in autophagy, and TRIM2 knockdown does not influence the level of autophagy, as evidenced by the detection of Beclin-1 and LC3-related proteins, which are involved in the initial steps of autophagy[19] (Supplementary Fig. 3F). Furthermore, we evaluated RIPK3-dependent necroptosis by assessing RIPK3 phosphorylation levels—a widely used biomarker for the instigation of necroptosis[20]—in intestinal tissues from WT and *Trim2*[-/-] mice subjected to II/R. Western blot analysis revealed that while II/R injury significantly increased RIPK3 phosphorylation compared to sham controls ($p < 0.05$), no statistically significant difference was observed between WT and *Trim2*[-/-] groups (Supplementary Fig. 3G). These findings suggest that TRIM2 deficiency exacerbates II/R-induced intestinal injury by amplifying mitochondrial dysfunction and apoptotic signaling, without influencing autophagy activation and necroptosis.

### Reducing TRIM2 expression promotes H/R-induced apoptosis in intestinal epithelial cells

To investigate the direct regulatory role of TRIM2 in intestinal epithelial cell survival, we established IEC-6 and Caco-2 cell lines with stable TRIM2 knockdown using lentiviral infection techniques. H/R triggered a significant increase in LDH release—a marker of membrane integrity loss[21]—in both cell lines (Fig. 3A). TRIM2 silencing exacerbated this effect, with LDH activity elevated in knockdown cells compared to controls (Fig. 3A). Consistent with enhanced cytotoxicity, CCK8 assay results revealed that the viability of TRIM2-knockdown IEC-6 and Caco-2 cells was markedly diminished in comparison to control cells following H/R treatment (Supplementary Fig. 4A). Western blot analysis was conducted to evaluate the expression of apoptotic proteins, including Bax, Bcl2, Bad, and cleaved Casp3. As shown in Fig. 3B and C, the Bax/Bcl2 ratio, along with the levels of pro-apoptotic proteins Bad and cleaved Casp3, was elevated following H/R treatment, with these changes being more pronounced in TRIM2-deficient cells. To provide further support for these findings, qRT-PCR analysis demonstrated a significant elevation in Bax and Bad mRNA levels and a marked reduction in Bcl2 mRNA in TRIM2-knockdown IEC-6 and Caco-2 cells (Fig. 3D, see also Supplementary Fig. 4B, C). Subsequently, flow cytometry confirmed that TRIM2 deficiency heightened apoptotic susceptibility (Fig. 3E, see also Supplementary Fig. 5). These data collectively establish TRIM2 as a critical suppressor of H/R-induced apoptosis in intestinal epithelial cells.

### Overexpression of TRIM2 restrains H/R-induced apoptosis in intestinal epithelial cells

Building on the observation that TRIM2 knockout in mice and reduced TRIM2 expression in IEC-6 and Caco-2 cells exacerbate apoptosis induced by II/R or H/R injury, we conducted further investigations to ascertain whether TRIM2 overexpression could protect intestinal epithelial cells from H/R injury. To achieve this, HA-TRIM2 plasmids were transfected into IEC-6 and Caco-2 cells to overexpress TRIM2 protein. The LDH assays revealed that the release of LDH into the supernatant was markedly reduced in the TRIM2-overexpressing cells in comparison to the controls (Fig. 4A). Similarly, CCK8 assays demonstrated that the viability of TRIM2-overexpressing IEC-6 and Caco-2 cells was markedly higher than that of cells in the HA-Vector group following H/R treatment (Supplementary

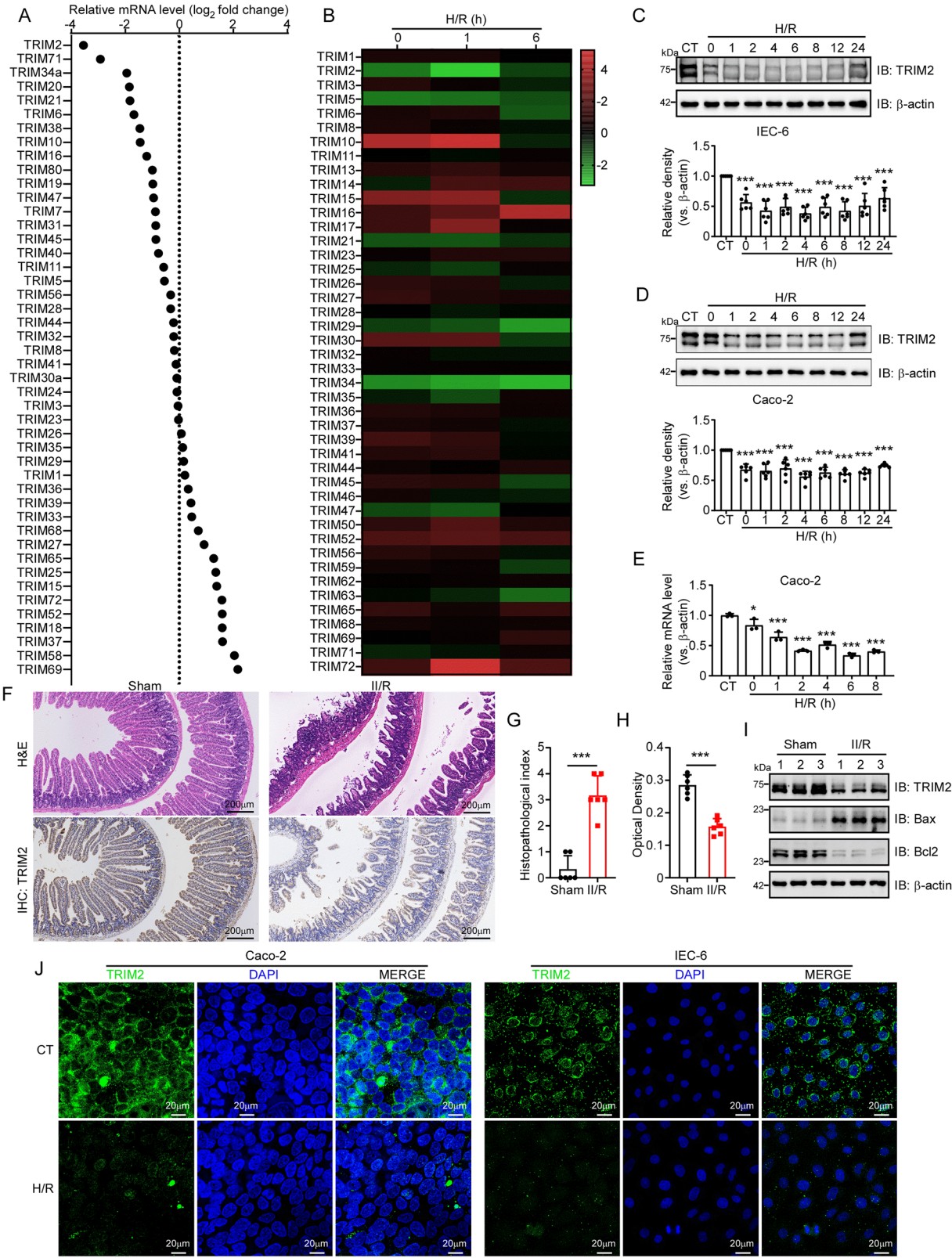

Fig. 6A). Mitochondrial membrane potential (MMP), a key indicator of early apoptosis[22], was assessed using JC-1 fluorescence. Following H/R treatment, a significant decline in MMP was observed in both the HA-Vector and HA-TRIM2 groups, while TRIM2 overexpression mitigated this decline (Fig. 4B, see also Supplementary Fig. 6B). Western blot analysis revealed that H/R upregulated pro-apoptotic proteins (Bax, Bad, and

cleaved Casp3) while suppressing anti-apoptotic protein Bcl2 in control cells (Fig. 4C, see also Supplementary Fig. 6C–F). However, TRIM2 over-expression reversed these effects, reducing Bax/Bcl2 ratio and cleaved Casp3 levels (Fig. 4C, see also Supplementary Fig. 6C–F). These findings were corroborated by flow cytometry analysis, which revealed a significant increase in the proportion of apoptotic cells after H/R treatment in IEC-6

**Fig. 1 | TRIM2 is downregulated in II/R injury models. A** qRT-PCR analysis of TRIM family mRNA expression in intestinal tissues of sham-operated mice and mice subjected to II/R injury. Data are normalized to the sham group. **B** TRIM family mRNA levels in IEC-6 cells exposed to hypoxia (24 h) followed by reoxygenation (0, 1, 6 h). Data are expressed relative to the untreated control (0 h). **C, D** Western blot analysis of TRIM2, Bax, and Bcl2 protein levels in Caco-2 (12 h hypoxia) and IEC-6 (24 h hypoxia) cells after reoxygenation. β-actin served as the loading control. n = 6 experimental repeats. **E** TRIM2 mRNA levels in Caco-2 cells under normoxia (CT) or H/R (n = 3 biologically independent samples). **F** Representative H&E and TRIM2 immunohistochemistry (IHC) staining of intestinal tissues from sham and II/R groups. Scale bars: 200 μm. **G** Intestinal injury severity quantified using Chiu's scoring system (*n* = 6 mice for each group). **H** TRIM2 IHC staining intensity analyzed via ImageJ (*n* = 6 mice for each group). **I** Western blot of TRIM2, Bax, and Bcl2 protein expression in intestinal tissues from sham and II/R mice. β-actin was used for normalization. **J** Immunofluorescence staining of TRIM2 (green) in Caco-2 and IEC-6 cells under CT or H/R conditions. Nuclei were counterstained with DAPI (blue). Scale bars: 20 μm. Data are presented as mean ± SD. *$p < 0.05$, ***$p < 0.001$ (Student's t-test).

and Caco-2 cells (Fig. 4D, see also Supplementary Fig. 7). Notably, the proportion of apoptotic cells was significantly lower in the HA-TRIM2 group compared to the HA-Vector group (Fig. 4D, see also Supplementary Fig. 7). In addition, we also validated the results in mouse IECs, which were consistent with the cell line results (Supplementary Fig. 8). These findings demonstrate that TRIM2 overexpression preserves mitochondrial integrity and suppresses apoptotic signaling, conferring protection against H/R injury in intestinal epithelial cells.

## TRIM2 directly interacts with BNIP3

To further elucidate the molecular regulatory mechanism of TRIM2 in II/R injury, we identified BNIP3 as a potential interacting protein of TRIM2 using the quantitative ubiquitinated proteome (Supplementary Fig. 9A). To confirm the interaction between TRIM2 and BNIP3, the HA-TRIM2 and Flag-BNIP3 plasmids were co-transfected into HEK293T cells. An exogenous interaction between TRIM2 and BNIP3 was confirmed by co-immunoprecipitation and western blot analysis (Fig. 5A and B). Additionally, H/R treatment amplified endogenous TRIM2-BNIP3 binding in Caco-2 cells (Fig. 5C), suggesting stress-dependent association. To observe the subcellular localization of TRIM2 and BNIP3, immunofluorescence experiments were conducted in Caco-2 and IEC-6 cells. The results demonstrated that TRIM2 is distributed in both the cytoplasm and the nucleus, with a predominantly cytoplasmic localization, whereas BNIP3 is exclusively localized to the cytoplasm. H/R treatment enhanced the co-localization of TRIM2 and BNIP3, which was consistent with the findings of the endogenous co-immunoprecipitation (Fig. 5D). It is worth noting that TRIM2 itself is partially localized on mitochondria, and H/R does not seem to have a significant effect on its localization (Supplementary Fig. 9B). To determine whether TRIM2 directly binds BNIP3, a GST pull-down assay was conducted. The purified GST-TRIM2 protein was observed to efficiently pull-down Flag-BNIP3 expressed in HEK293T cells, thereby confirming a direct interaction between these proteins (Fig. 5E).

The next aim was to identify the structural domains required for the TRIM2-BNIP3 interaction. To this end, various GFP-TRIM2 deletion mutants were generated, including those comprising the Ring finger domain, two B-box-type zinc finger (BB2) domains, a coiled-coil (CC) domain, a filamin-type IG (fn) domain, and the NCL-1, HT2A, and Lin-41 (NHL) domain, were generated (Fig. 5G). Flag-BNIP3 and GFP-TRIM2 deletion mutants were co-transfected into HEK293T cells, and the resulting protein complexes were subjected to co-immunoprecipitation and western blot analysis. Interactions with Flag-BNIP3 were observed in all GFP-TRIM2 domains with the exception of the Ring finger domain (Fig. 5F). Additionally, in order to identify the region of BNIP3 required for interaction with TRIM2, various Flag-BNIP3 deletion mutants were generated (Fig. 5J). The aforementioned mutants, in conjunction with the GFP-TRIM2 construct, were transfected into HEK293T cells for subsequent co-immunoprecipitation and western blot analysis. The results demonstrated that the C-terminal region of BNIP3 (amino acids 164–194) is essential for its interaction with TRIM2 (Fig. 5H). This finding was further validated by GST pull-down assays (Fig. 5I, see also Supplementary Fig. 10). Studies have shown that BNIP3 can selectively bind to Bcl2 and Bax to regulate mitochondrial apoptosis[23]. We also confirmed through immunoprecipitation that BNIP3 can bind to Bcl2 and Bax, but cannot bind to Bad

(Supplementary Fig. 11). Taken together, these results confirm that TRIM2 directly interacts with BNIP3 in intestinal epithelial cells.

## TRIM2 facilitates K48-linked polyubiquitination of BNIP3

As an E3 ubiquitin ligase, TRIM2 catalyzes substrate ubiquitination to regulate biological processes[24]. Co-transfection of Flag-BNIP3 and HA-Ub in HEK293T cells revealed that TRIM2 markedly augmented BNIP3 ubiquitination (Fig. 6A). To define the ubiquitin chain topology, we employed lysine-restricted ubiquitin mutants (K6O, K11O, etc.). TRIM2 selectively promoted K48-linked polyubiquitination of BNIP3, as evidenced by preserved ubiquitination only with WT or K48O ubiquitin (Fig. 6B). This specificity was further confirmed using K48R ubiquitin mutants, which abolished TRIM2-mediated BNIP3 ubiquitination (Supplementary Fig. 12). Immunoblotting with K48- and K63-specific antibodies corroborated exclusive K48 linkage (Fig. 6C). Co-expression of HA-K48R-Ub with TRIM2 and BNIP3 in HEK293T cells abolished ubiquitination, confirming K48 as the critical residue (Fig. 6D).

In Caco-2 cells, TRIM2 overexpression increased K48-linked ubiquitination of BNIP3 (Fig. 6E), while TRIM2 knockdown reduced it (Fig. 6F). To identify ubiquitination sits on BNIP3, lysine-to-arginine mutants were generated (Fig. 6G). Ubiquitination assays revealed a reduction in polyubiquitination for the BNIP3 K130R mutant, pinpointing K130 as the primary site for TRIM2-mediated modification (Fig. 6H). In conclusion, TRIM2 acts as an E3 ligase to catalyze K48-linked polyubiquitination of BNIP3, with lysine 130 serving as the critical acceptor site.

## TRIM2 accelerates proteasomal degradation of BNIP3

Consistent with K48-linked ubiquitination's role in proteasomal degradation[25], overexpression of TRIM2 in Caco-2 cells significantly reduced BNIP3 protein levels without altering *BNIP3* mRNA (Fig. 7B). This effect was dose-dependent, with BNIP3 levels declining progressively as TRIM2 expression increased (Fig. 7C and D). To investigate the degradation pathway of BNIP3 mediated by TRIM2, Caco-2 cells transfected with HA-TRIM2 plasmids were treated with DMSO (control), MG132 (proteasome inhibitor), chloroquine (CQ, lysosome inhibitor), or 3-MA (autophagy inhibitor) for 6 hours. MG132 treatment reversed the TRIM2-mediated degradation of BNIP3 (Fig. 7E), indicating the involvement of the proteasome. Furthermore, the stabilization of BNIP3 protein by MG132 was time-dependent (Fig. 7F). We next examined the effect of TRIM2 on the half-life of BNIP3 degradation in Caco-2 cells using cycloheximide (CHX), an inhibitor of protein synthesis. In TRIM2-overexpressing cells, the degradation half-life of BNIP3 was significantly shortened, and this effect was completely inhibited by MG132 (Fig. 7G). Conversely, knockdown of TRIM2 significantly prolonged the degradation half-life of BNIP3, and this effect was reversed upon reintroduction of TRIM2 (Fig. 7H). Given our previous findings that lysine 130 (K130) is a critical site for TRIM2-mediated BNIP3 polyubiquitination, we co-transfected HEK293T cells with Flag-BNIP3-WT or Flag-BNIP3-K130R mutant plasmids together with HA-TRIM2. The results showed that TRIM2 significantly shortened the degradation half-life of wild-type BNIP3, but had no effect on the K130R mutant (Fig. 7I). Moreover, the colocalization of TRIM2 and Flag-BNIP3 in Caco-2 cells treated with MG132 was reduced when Flag-BNIP3 was mutated to K130R (Fig. 7J). Collectively, TRIM2 promotes proteasomal

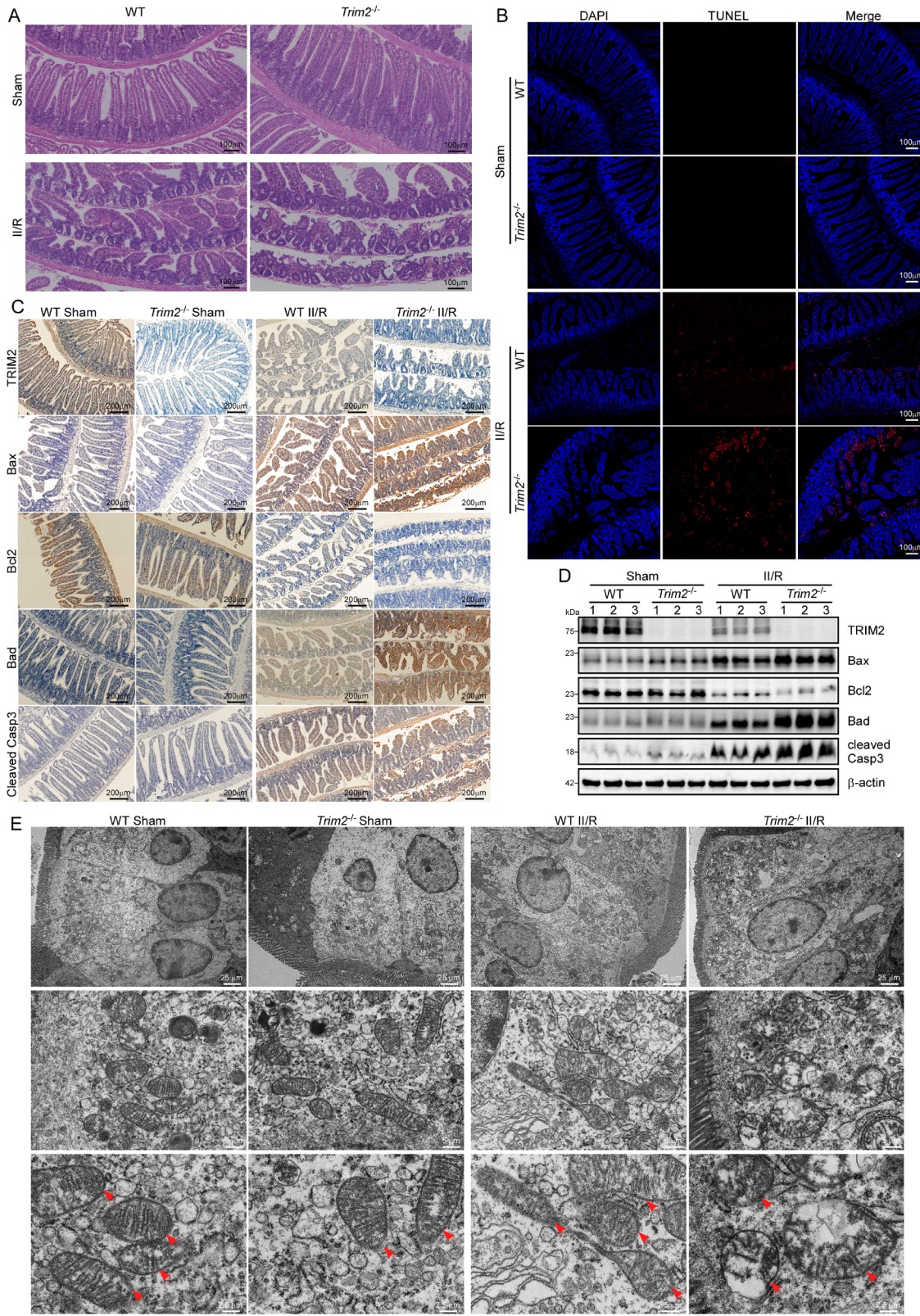

**Fig. 2 | The Knockout of TRIM2 exacerbates II/R-induced intestinal apoptosis.**
**A** Representative H&E-stained intestinal sections from WT mice and *Trim2⁻ᐟ⁻* mice under sham or II/R conditions (*n* = 6 per group). All images are presented at a scale of 100 μm. **B** Representative images for TUNEL staining of the intestine in the four groups of mice (n = 6 per group), with a scale bar of 100 μm. **C** Representative photomicrographs of immunohistochemical staining of the intestine in four groups of mice using antibodies against TRIM2, Bax, Bcl2, Bad, and cleaved Casp3 (*n* = 6 per group). The scale bars are 200 μm. **D** The protein expressions of TRIM2, Bax, Bcl2, BAD, and cleaved Casp3 in the four groups of mice were detected by Western blot in the intestine. β-actin was served as a loading control. (E) The morphological changes of mitochondria in the intestinal tissues of the four groups of mice were observed by TEM. Scale bars: 25 μm (overview), 5 μm (mid-magnification), 2.5 μm (high-magnification).

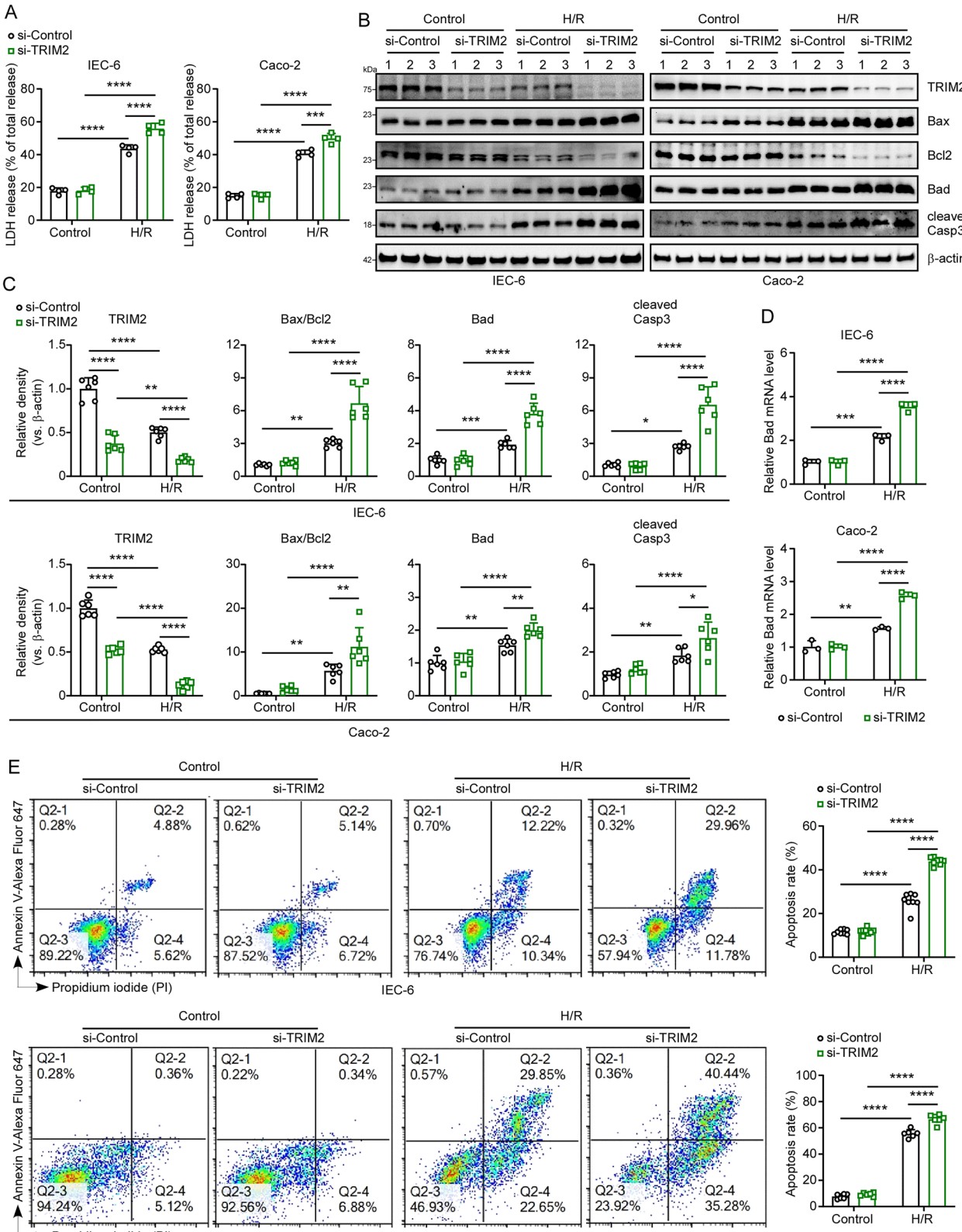

**Fig. 3 | Reducing TRIM2 expression promotes H/R-induced apoptosis in intestinal epithelial cells. A** Lactate dehydrogenase (LDH) activity in the supernatants of IEC-6 and Caco-2 cells following H/R treatment (n = 4 biologically independent samples). **B** The protein expression levels of TRIM2, Bax, Bcl2, Bad and cleaved Casp3 in IEC-6 and Caco-2 TRIM2-knockdown cells were determined by Western blotting following H/R treatment. β-actin was used as a loading control. **C** Quantification of the TRIM2, Bax, Bcl2, Bad and cleaved Casp3 western blot bands (n = 6 biologically independent samples). **D** BAD mRNA levels in IEC-6 and Caco-2 TRIM2-knockdown cells subjected to control or H/R treatment (n = 3 biologically independent samples). **E** Flow cytometry was employed to ascertain the extent of apoptosis in IEC-6 (n = 8 biologically independent samples) and Caco-2 (n = 6 biologically independent samples) TRIM2-knockdown cells subjected to control or H/R treatment. All results are expressed as the mean ± SD. Statistical significance was determined using one-way ANOVA followed by Tukey's test, with the following levels of significance: $*p < 0.05$, $**p < 0.01$, $***p < 0.001$, $****p < 0.0001$.

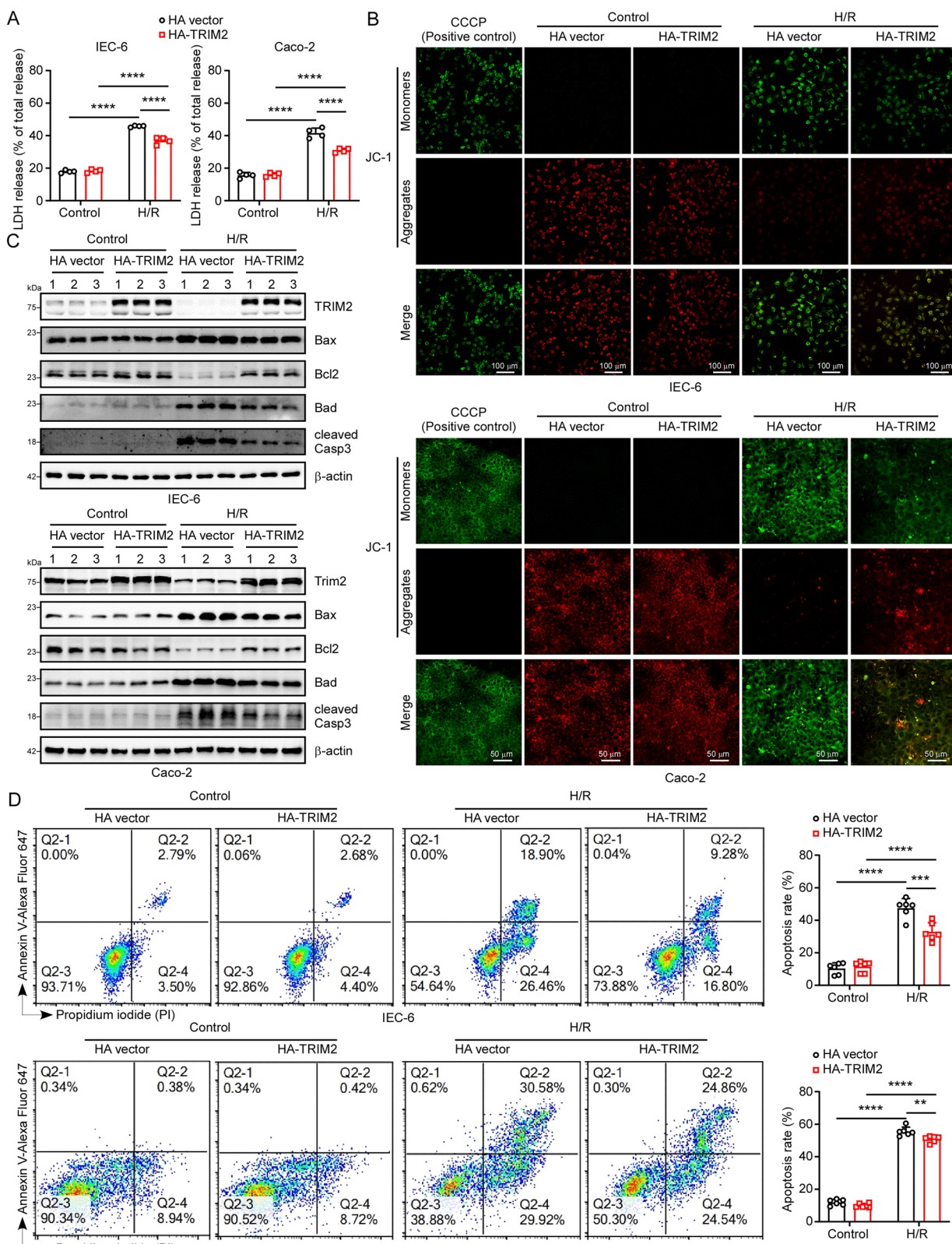

**Fig. 4 | Overexpression of TRIM2 restrains H/R-induced apoptosis in intestinal epithelial cells. A** LDH activity in IEC-6 and Caco-2 cell supernatants following H/R treatment ($n = 4$ biologically independent samples). **B** Following transfection with either the HA-Vector or the HA-TRIM2 plasmid, IEC-6 and Caco-2 cells were stained with JC-1 for a period of 20 minutes at 37 °C. The cells were then observed using a laser scanning confocal microscope. Scale bars are 100 μm (IEC-6) and 50 μm (Caco-2). **C** The protein expression levels of TRIM2, BAX, Bcl2, BAD and cleaved Casp3 in IEC-6 and Caco-2 TRIM2-overexpression cells were determined by

Western blot following H/R treatment. β-actin was used as a loading control. **D** Flow cytometry was employed to ascertain the extent of apoptosis in IEC-6 and Caco-2 TRIM2-overexpression cells subjected to control or H/R treatment ($n = 6$ biologically independent samples). All results are expressed as the mean ± SD. Statistical significance was determined using one-way ANOVA followed by Tukey's test, with the following levels of significance: *$p < 0.05$, **$p < 0.01$, ***$p < 0.001$, ****$p < 0.0001$.

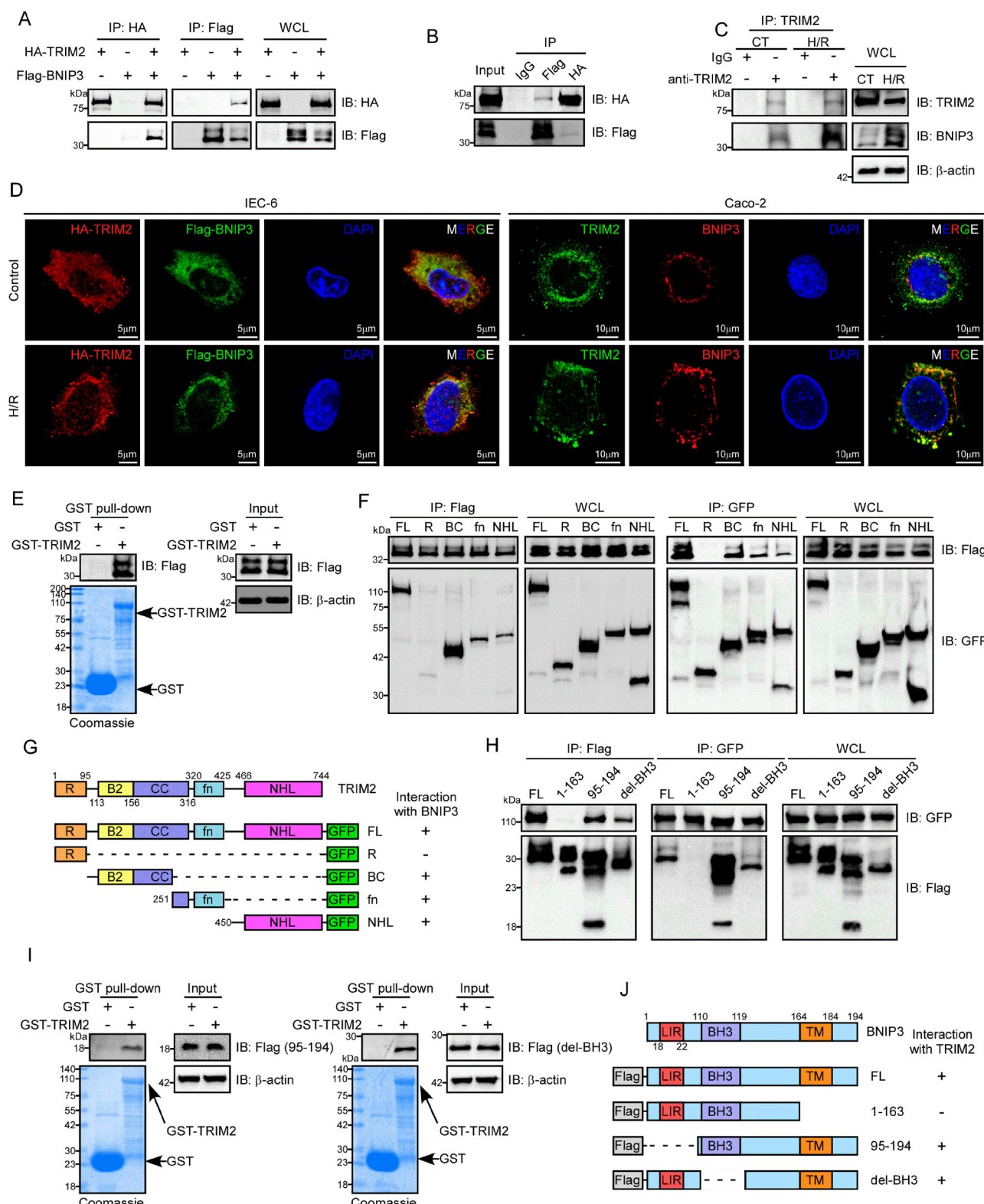

degradation of BNIP3 by catalyzing K48-linked ubiquitination at lysine 130, a process critical for regulating BNIP3 stability in intestinal epithelial cells.

## TRIM2 targets BNIP3 to inhibit II/R-induced apoptosis

To determine whether BNIP3 functionally intersects with TRIM2 in regulation of apoptosis, Caco-2 cells were co-transfected with small interfering RNAs (siRNAs) targeting TRIM2 and BNIP3 to inhibit their protein expression. TRIM2 knockdown exacerbated H/R-induced LDH release, while simultaneous BNIP3 knockdown attenuated this effect (Fig. 8A). Correspondingly, BNIP3 silencing restored cell viability in TRIM2-deficient cells (Fig. 8B). Flow cytometry analysis further supported these findings, showing that TRIM2 deficiency significantly increased the proportion of apoptotic cells in H/R-treated Caco-2 cells. However, this effect was reversed by BNIP3 knockdown (Fig. 8C, D, see also Supplementary Fig. 13A).

**Fig. 5 | TRIM2 directly interacts with BNIP3. A**, **B** The HA-tagged TRIM2 and Flag-tagged BNIP3 plasmids were co-transfected into HEK293T cells. The exogenous interaction between TRIM2 and BNIP3 was confirmed by co-immunoprecipitation and western blot analysis. **C** The endogenous interaction between TRIM2 and BNIP3 was detected by co-immunoprecipitation and western blot in Caco-2 cells subjected to CT or H/R treatment. **D** Representative confocal images of Caco-2 and IEC-6 cells demonstrated the co-localization and distribution of TRIM2 and BNIP3. **E** Flag-tagged BNIP3 plasmids were transfected into HEK293T cells, and lysates were collected from the HEK293T cells after 24 hours of transfection. The purified GST or GST-TRIM2 proteins were incubated with the collected lysates overnight. The direct interaction between TRIM2 and BNIP3 was verified by a GST-pulldown and western blot experiment, and the purity of GST-TRIM2 was assessed through Coomassie blue staining. **F** Full-length Flag-BNIP3 and a series of deletion mutants of GFP-TRIM2 were transfected into HEK293T cells. The interaction was verified by co-immunoprecipitation and western blot analysis. **G** A schematic representation of the plasmid generation for various deletion mutants of GFP-TRIM2 and their interaction with BNIP3. **H** Full-length GFP-TRIM2 and various deletion mutants of Flag-BNIP3 were co-transfected into HEK293T cells. The interaction between the two proteins was confirmed through co-immunoprecipitation and western blot analysis. **I** Various deletion mutants of Flag-tagged BNIP3 plasmids were transfected into HEK293T cells, and lysates were collected from the HEK293T cells after 24 hours of transfection. The purified GST or GST-TRIM2 proteins were incubated with the collected lysates overnight. The direct interaction between TRIM2 and the deletion mutants of Flag-BNIP3 was verified by GST-pull down and western blot experiment, and the purity of GST-TRIM2 was assessed through Coomassie blue staining. **J** Schematic representation of plasmid generation for various deletion mutants of Flag-BNIP3 and the interaction with TRIM2.

Western blot analysis of apoptotic markers revealed that BNIP3 knockdown reduced the levels of pro-apoptotic proteins (Bax, Bad, cleaved Casp3) and increased the level of the anti-apoptotic protein (Bcl2) in TRIM2-knockdown cells (Fig. 8E, see also Supplementary Fig. 14A). Moreover, overexpression of BNIP3 was found to counteract the protective effects of TRIM2 against H/R-induced injury, as indicated by LDH and CCK8 assay results (Figs. 8F and 8G). Flow cytometry experiments confirmed that excess TRIM2 reduced H/R-induced apoptosis, but this protective effect was attenuated by BNIP3 overexpression (Fig. 8H, see also Supplementary Fig. 13B). Consistent with these findings, western blot analysis showed that BNIP3 overexpression increased levels of pro-apoptotic proteins (Bax, Bad, cleaved Casp3) and decreased Bcl2 levels, even in TRIM2-overexpressing cells (Fig. 8I, see also Supplementary Fig. 14B).

To assess MMP, a JC-1 assay was performed in Caco-2 cells co-transfected with HA-TRIM2 and Flag-BNIP3 plasmids. The results revealed that the increase in MMP induced by TRIM2 overexpression was abolished when BNIP3 was also overexpressed (Fig. 8J). Finally, to demonstrate that TRIM2 mediates apoptosis by promoting the ubiquitination and degradation of BNIP3, Caco-2 cells were co-transfected with HA-TRIM2 and either Flag-BNIP3-WT or Flag-BNIP3-K130R plasmids prior to H/R treatment. The LDH assay showed that HA-TRIM2 significantly reduced LDH release in Flag-BNIP3-WT cells but had no effect in Flag-BNIP3-K130R cells, in which BNIP3 cannot be ubiquitinated by TRIM2 (Fig. 8K). Similarly, the CCK8 assay indicated that TRIM2 improved cell viability in Flag-BNIP3-WT cells, but this protective effect was abolished in Flag-BNIP3-K130R cells (Fig. 8L). Consistent with previous findings, TRIM2 significantly inhibited H/R-induced apoptosis in Flag-BNIP3-WT cells, but failed to produce a similar effect in Flag-BNIP3-K130R cells (Supplementary Fig. 14 C). In conclusion, TRIM2 regulates apoptosis in intestinal epithelial cells by targeting BNIP3 for ubiquitination and subsequent proteasomal degradation.

## Discussion

The ubiquitin-proteasome system (UPS)—comprising ubiquitin (Ub), Ub-activating enzyme (E1), Ub-conjugating enzyme (E2), ubiquitin ligase (E3), deubiquitinating enzyme (DUB), and the 26S proteasome—serves as a central regulatory mechanism in diverse pathophysiological processes, from protein quality control to apoptotic signaling[26]. Within this system, E3 ubiquitin ligases are pivotal in determining the specific substrate proteins targeted for degradation through ubiquitination[27]. The tripartite motif (TRIM) protein family, one of the largest classes of RING-type E3 ubiquitin ligases, has garnered increasing attention for its multifaceted roles in apoptosis and cellular stress responses. For instance, TRIM72, also known as MG53, participates in repairing cell membrane damage and exerts protective effects against ischemia/reperfusion (I/R) injury in multiple oxygen-dependent organs, including the heart, brain, lungs, kidneys, and liver[28]. TRIM27 alleviates liver I/R injury by suppressing apoptosis and inflammatory responses[29], while TRIM47 modulates apoptosis and inflammation via activation of nuclear factor-kappa B (NF-κB) signaling in cerebral I/R injury[30]. These findings underscore the divergent yet critical roles of TRIM proteins in modulating cell survival under ischemic stress.

Recently, we identified TRIM65 as a key regulator that reduces II/R-induced apoptosis through the ubiquitination of Tox4, with TRIM65 deficiency exacerbating apoptosis[31]. Despite these findings, the roles and underlying mechanisms of other TRIM proteins in II/R injury remains largely uncharted. To address this gap, we systematically screened TRIM family members in murine II/R models and H/R-treated IEC-6 cells, identifying TRIM2 as the most dynamically regulated candidate. Subsequent validation across in vivo and in vitro models confirmed consistent downregulation of TRIM2 post-II/R, aligning with its proposed role as a stress-responsive regulator. Emerging evidence highlights TRIM2's transcription as being governed by a multifactorial regulatory network, including transcription factors (e.g., p53 and C/EBPβ, which directly bind its promoter)[32], post-transcriptional modifiers (e.g., miRNAs such as miR-145, miR-181c, and miR-222-5p that target TRIM2's 3'UTR)[33–37], and competing endogenous RNAs (e.g., circORC2, LINC01535, and NR2F1-AS1 that counteract miRNA-mediated repression)[38–40]. These findings underscore the complexity of TRIM2 regulation, which may integrate stress-responsive pathways (e.g., GPER/MAPK/ERK)[41] and epigenetic modifiers. Based on these findings, we hypothesize that TRIM2 may play a critical role in the pathophysiology of II/R injury, and its expression regulation mechanism in II/R deserves further exploration.

TRIM2 is an 81 kDa multidomain protein comprising a RING finger (R) domain, two B-box-type zinc finger (B2) domains, a coiled-coil (CC) domain, a filamin-type IG (fn) domain, and an NCL-1, HT2A, and Lin-41 (NHL) domain[24]. Previous studies have implicated that TRIM2 plays roles in a variety of physiological and pathological processes, including neuronal rapid ischemic tolerance, antiviral responses, neurological diseases, tumor proliferation, migration, invasion, and apoptosis[24]. In particular, TRIM2 has been reported to bind to the Bcl2-interacting mediator of cell death (Bim) and mediate its degradation through ubiquitination, thereby contributing to neuroprotection in rapid ischemic tolerance[15]. Additionally, recent research has demonstrated that TRIM2 selectively promotes inflammation-driven pathological angiogenesis by modulating the induction of critical angiogenic mediators and the phosphorylation of eNOS[42]. Despite these insights, the role of TRIM2 in II/R injury remains unexplored. To address this gap, we employed TRIM2 knockout mice and two types of intestinal epithelial cells to establish II/R and H/R injury models. In vitro, overexpression of TRIM2 in IEC-6 and Caco-2 cells significantly reduced apoptosis following H/R treatment, whereas TRIM2 knockdown increased apoptosis. In vivo, TRIM2 deficiency in mice markedly exacerbated intestinal apoptosis after II/R injury. Based on the findings of this study, we conclude that TRIM2 plays a critical role in regulating apoptosis during II/R injury, highlighting its potential as a promising therapeutic target for mitigating II/R injury.

To explore the underlying mechanism, we identified BNIP3 as a substrate of TRIM2. BNIP3, localized at the mitochondrial outer membrane, has been found to function as a positive regulator of mitochondrial autophagy through its interaction with LC3 on autophagosomes via its N-terminal LC3-interacting region[43]. However, previous studies have also shown that BNIP3 can act as a pro-apoptotic protein and contribute to the pathogenesis of various diseases by inducing apoptosis and mitochondrial

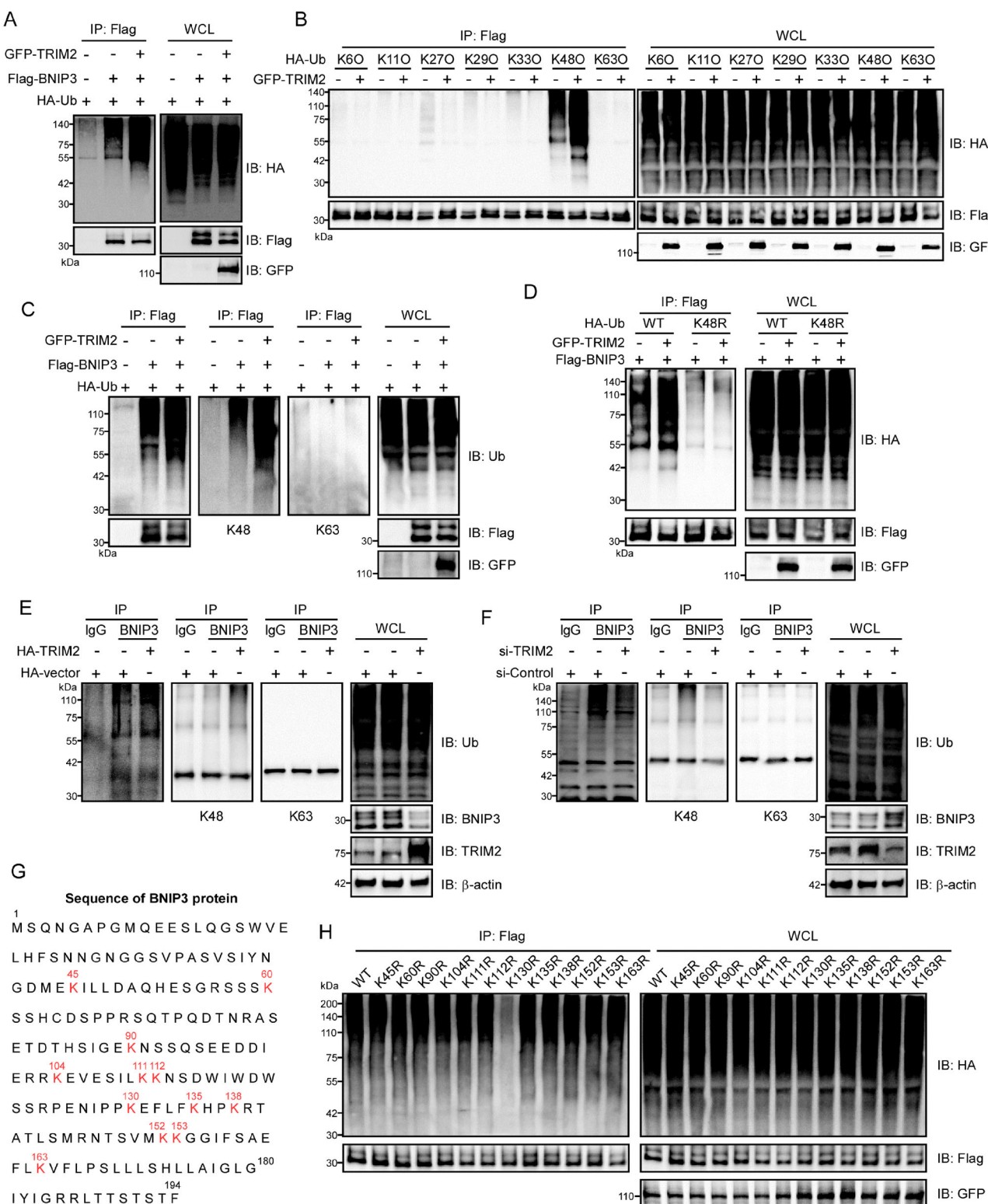

dysfunction[44–48]. To clarify the relationship between TRIM2 and BNIP3, we utilized multiple experimental approaches, including co-IP, IF, and GST pull-down assays, to demonstrate a direct interaction between TRIM2 and BNIP3. Furthermore, as an E3 ubiquitin ligase, TRIM2 was shown to promote K48-linked polyubiquitination of BNIP3 at lysine 130 (K130). As a BH3-only protein within the Bcl2 family, BNIP3 effectively activates downstream effectors Bax/Bak[46,49], leading to the opening of the mitochondrial permeability transition pore (mPTP), ultimately resulting in

apoptosis and mitochondrial dysfunction[47,50,51]. Interestingly, it was found that the transmembrane (TM) domain of BNIP3, which interacts with TRIM2, was found to play a critical role in the induction of apoptosis, whereas its BH3 domain is less relevant to this process. Notably, the TM domain of BNIP3 is not essential for its role in mitophagy[52,53]. In this study, we also observed that upregulation of BNIP3 in intestinal epithelial cells abolished the protective effects of TRIM2 against apoptosis induced by H/R treatment, whereas downregulation of BNIP3 restored these protective

**Fig. 6 | TRIM2 facilitates K48-linked polyubiquitination of BNIP3. A** Flag-BNIP3 and HA-Ub plasmids were co-transfected into HEK293T cells in the presence or absence of TRIM2. Polyubiquitination of BNIP3 was quantified by co-immunoprecipitation and western blot analysis. **B** The indicated plasmids were co-transfected into HEK293T cells for 24 hours. The absence and level of ubiquitination of BNIP3 were then detected by co-immunoprecipitation with anti-Flag tag antibodies and immunoblotting (IB) with the indicated antibodies. **C** Flag-BNIP3 and HA-Ub plasmids were co-transfected into HEK293T cells for 24 hours, with or without TRIM2. Polyubiquitination of BNIP3 was detected by IB with a specific antibody for Ub, K48 Ub, or K63 Ub after immunoprecipitation. **D** HEK293T cells were co-transfected with the indicated plasmids for 24 hours. The absence and level of ubiquitination of BNIP3 were then detected by co-immunoprecipitation with anti-Flag tag antibodies, and IB with the indicated antibodies. **E** The endogenous ubiquitination of BNIP3 in Caco-2 cells transfected with HA-TRIM2 or HA-Vector plasmids was detected by IB with the specific antibody for Ub, K48, or K63 after immunoprecipitation. **F** The endogenous BNIP3 ubiquitination in Caco-2 cells infected with Lenti-si-TRIM2 or Lenti-si-Control virus was detected by IB with the specific antibody for Ub, K48, or K63 after immunoprecipitation. **G** A graphical representation of the 12-point mutation site in BNIP3, in which all lysine residues (K) were replaced with arginine (R). **H** HEK293T cells were co-transfected with the indicated plasmids for 24 hours. The polyubiquitination sites of BNIP3 by TRIM2 were identified in the lysates of HEK293T cells by IB with an anti-HA antibody after immunoprecipitation with anti-Flag tag antibodies.

effects. These findings suggest that BNIP3 is a substrate of TRIM2 and plays an important role in II/R injury.

Recent studies have revealed that the UPS plays a pivotal role in BNIP3 degradation[48]. For instance, a ubiquitin ligase complex composed of SKP1, CUL1, and an F-box protein has been shown to mediate the ubiquitination and degradation of BNIP3[54]. In the present study, we found that overexpression of TRIM2 significantly promotes the ubiquitination and degradation of BNIP3, whereas TRIM2 deficiency results in deubiquitination and stabilization of BNIP3. The process of substrate ubiquitination involves the addition of a single ubiquitin molecule or the formation of polyubiquitin chains[55]. Each of the seven lysine residues in ubiquitin (K6, K11, K27, K29, K33, K48, and K63) can form linkages that result in structurally distinct polyubiquitin chains. These linkages interact with specific substrates to produce distinct functional outcomes. For instance, K48-linked polyubiquitination is well-established as a signal for proteasomal degradation[9]. In this study, we systematically screened the seven types of polyubiquitin linkages (K6, K11, K27, K29, K33, K48, and K63) using ubiquitination assays. The results demonstrated that TRIM2 mediates the K48-linked ubiquitination of BNIP3. Consistent with previous findings, this K48-linked ubiquitination of BNIP3 is associated with the ubiquitin-proteasome degradation pathway. These findings provide mechanistic insight into the protective role of TRIM2 in reducing II/R-induced apoptosis and further highlight the importance of TRIM2-mediated regulation of BNIP3 in intestinal ischemia/reperfusion injury.

This study establishes TRIM2 as a key suppressor of II/R-induced apoptosis, operating through K48-linked ubiquitination and degradation of BNIP3. The discovery of this axis not only elucidates a previously unrecognized mechanism in II/R pathology but also positions TRIM2 as a promising therapeutic target. Future studies exploring TRIM2's regulatory network or pharmacological enhancers of its activity could unlock strategies for mitigating intestinal ischemia/reperfusion injury.

## Methods

### Animals
We have complied with all relevant ethical regulations for animal use. All experimental protocols involving animals were approved by the Ethics Committee of The First Affiliated Hospital of Nanchang University (Approval No. CDYFY-IACUC-202407QR215) and conducted in accordance with the *Guide for the Care and Use of Laboratory Animals* (NIH publication No. 86-23, revised 2011). Mice were housed in individually ventilated squirrel cages under pathogen-free conditions, with *ad libitum* access to food and water for the duration of the study. The TRIM2 knockout (*Trim2^-/-*) mice (strain number T052403) on a C57BL/6 J background were generated by GemPharmatech (Nanjing, China) using the CRISPR/Cas9 technology. A schematic of the *Trim2^-/-* mouse generation process is provided in Fig. S2. Homozygous offspring were identified by polymerase chain reaction (PCR) using the following primer pairs: Forward 1: AGTGCT-GAAGTCCATAGATCGG; Reverse 1: CTGATTCTCCTCATCACCAGG; Forward 2: AGTGCTGAAGTCCATAGATCGG; Reverse 2: GTCACT-CACTGCTCCCCTGT), and the age-matched homozygous *Trim2^-/-* and wild-type (WT) male mice (20-25 g) were used to establish II/R models.

### Mouse II/R model
Due to the protective effect of estrogen in II/R injury[56,57], only male mice were used in this study. Male C57BL/6 J mice (8 weeks) underwent a 12-hour fasting period with unrestricted water access prior to procedures. Mice were anesthetized via inhalation of 3% isoflurane for induction and maintained under 1.5% isoflurane in oxygen. A midline laparotomy was performed to expose the superior mesenteric artery (SMA), which was occluded with a microvascular clamp for 60 minutes. Successful ischemia was confirmed by intestinal dark red discoloration, loss of SMA pulsation, and absence of peristaltic activity. Reperfusion was initiated by clamp removal, restoring blood flow for 6 hours. Sham-operated mice underwent identical surgical steps, excluding SMA occlusion. The abdomen was then closed, and buprenorphine (30 μg/kg) was administered subcutaneously before suturing the wound. Following 6 hours of reperfusion, all mice were euthanized via isoflurane overdose. Intestinal tissue was promptly collected for subsequent analyses.

### Reagents and antibodies
Chloroquine (CQ) (HY-17589A, CAS: 54-05-7), MG-132 (HY-13259, CAS: 133407-82-6), Cycloheximide (CHX) (HY-12320, CAS: 66-81-9) and 3-Methyladenine (3-MA) (HY-19312, CAS: 5142-23-4) were purchased from MedChemExpress (MCE). The antibodies used were anti-TRIM2 (20356-1-AP and 67342-1-Ig), anti-Bax (50599-2-Ig), anti-Bcl2 (12789-1-AP), anti-cleaved-Casp3 (19677-1-AP), anti-Bad (10435-1-AP), anti-β-actin (67735-1-Ig), anti-Flag (66008-4-Ig and 20543-1-AP), anti-GFP (66002-1-Ig and 50430-2-AP), and anti-HA (66006-2-Ig and 51064-2-AP) antibodies purchased from Proteintech (Wuhan, China). The Ubiquitin (ab7780), K48 ubiquitin (ab140601), and K63 ubiquitin (ab179434) antibodies were purchased from Abcam. The BNIP3 antibody (44060S) was purchased from Cell Signaling Technology. The p-Ripk3 (AF7443) and Ripk3 (AF7942) antibody were purchased from Affinity Biosciences, siR-NAs targeting Binp3 were purchased from Tsingke Technology (Beijing, China). The Cell Count Kit-8 (CCK-8) and the ECL chemiluminescent substrate kit were purchased from ShareBio (Shanghai, China). The plasmid extraction kit was obtained from TIANGEN Technology (Beijing, China). The Protein A + G (Fast) was purchased from Beyotime (Shanghai, China). The Polyethylenimine Linear (PEI) MW40000 was purchased from YEA-SEN (Shanghai, China).

### Cell culture and treatment
Caco-2 cells were purchased from Wuhan Pricella Biotechnology Co., Ltd., and cultured in MEM medium (Pricella, PM150410) supplemented with 20% fetal bovine serum (FBS; CellMax, SA101.02) and 1% penicillin/streptomycin (P/S; Servicebio, G4003). The HEK293T cells were obtained from ATCC, while the IEC-6 cells were sourced from Shanghai Anwei Biotechnology Co., Ltd. The HEK293T and IEC-6 cells were maintained in DMEM medium (Servicebio, G4511), supplemented with 10% FBS and 1% P/S, in a humidified incubator at 37 °C and 5% $CO_2$ in accordance with standard practice. Primary intestinal epithelial cells (IECs) from mouse intestine were obtained from Meisencell Biological Co., Ltd (CTCC-D007-MIC, Zhejiang, China). IECs were cultured in DMEM medium with epithelial cell culture additive, 10% fetal bovine

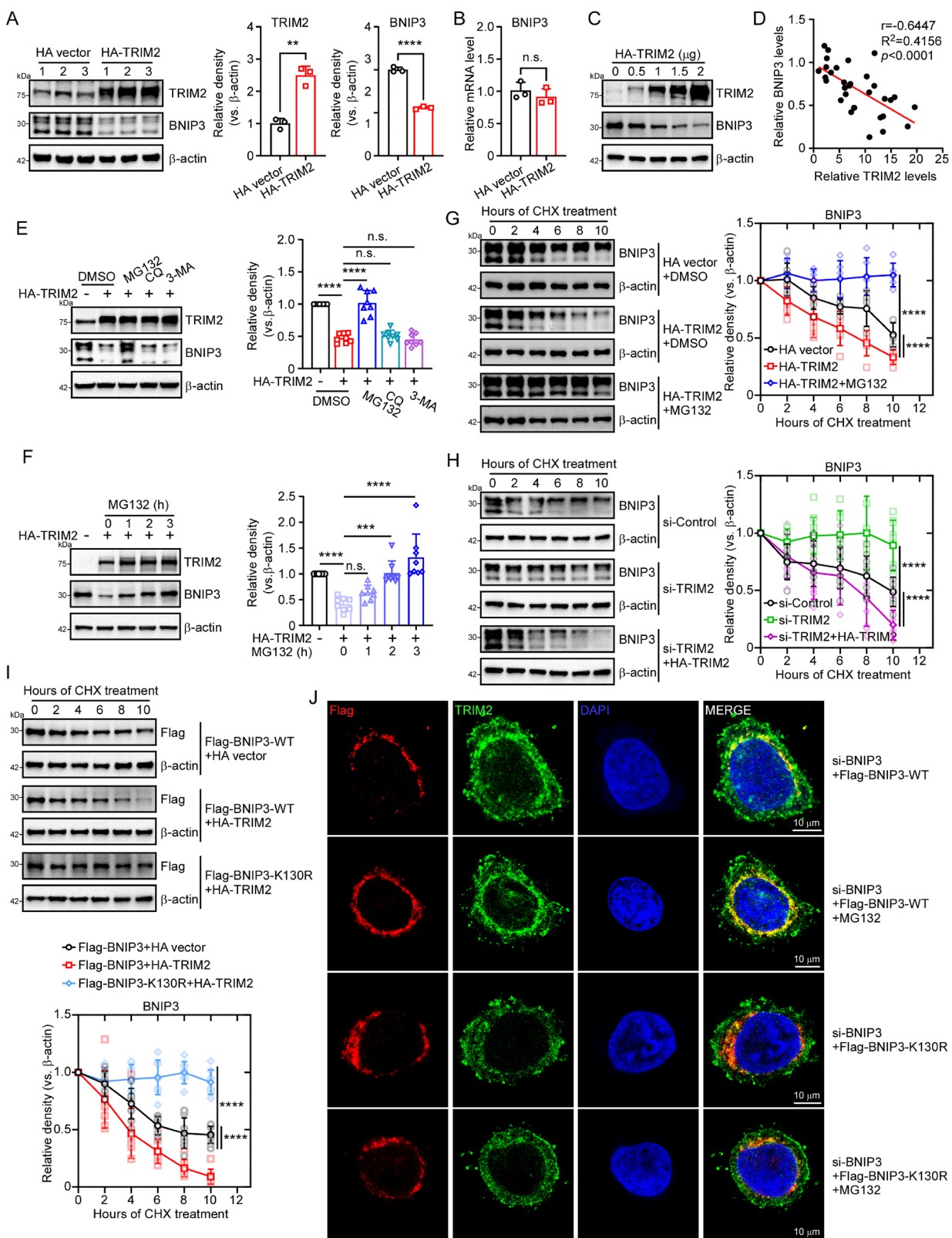

serum, and 1% penicillium streptomycin (CTCC-009-613, Zhejiang, China). All Cells were incubated at 37 °C in a humidified atmosphere with 5% $CO_2$. The hypoxia/reoxygenation (H/R) injury model was established as previously described[31]. Specifically, Cells were incubated in serum-free DMEM under hypoxic conditions (1% $O_2$, 5% $CO_2$, 94% $N_2$) for 24 hours (IEC-6) or 12 hours (Caco-3). Following this, the cells were

transferred to fresh medium containing 10% FBS and returned to normoxic conditions (21% $O_2$, 5% $CO_2$) for specified durations.

### Cell viability assay

Cell viability was evaluated using the Cell Counting Kit (CCK-8; SB-CCK8, Share-bio, Shanghai, China). IEC-6 or Caco-2 cells were seeded in 96-well

**Fig. 7 | TRIM2 accelerates proteasomal degradation of BNIP3. A** Caco-2 cells were transfected with either the HA-Vector or HA-TRIM2 plasmid using PEI transfection reagent. The protein expression of TRIM2 and BNIP3 was then detected by western blot. β-actin was used as a loading control (n = 3 biologically independent samples). **B** Caco-2 cells were transfected with HA-Vector or HA-TRIM2 plasmids using PEI transfection reagent. The relative mRNA level of BNIP3 in the two groups was then detected by qRT-PCR (n = 3 biologically independent samples). **C** HEK293T cells transfected with varying doses of HA-TRIM2 plasmids were harvested for western blot. β-actin was used as a loading control. **D** A correlation analysis was conducted to determine the relationship between the protein levels of TRIM2 and BNIP3 in cells (*n* = 32 biologically independent samples). **(E)** Caco-2 cells transfected with HA-TRIM2 plasmids were treated with dimethyl sulfoxide (DMSO), MG132 (a proteasome inhibitor), chloroquine (CQ, a lysosome inhibitor), and 3-methyladenine (3-MA, an autophagy inhibitor) for six hours. Western blot was employed to detect the protein expression levels of TRIM2 and BNIP3. The BNIP3 western blot bands were quantified in the different groups (*n* = 8 biologically independent samples). β-actin served as a loading control. **F** HEK293T cells transfected with HA-TRIM2 plasmids were treated with MG132 for 0 h, 1 h, 2 h and 3 h. Western blot was employed to evaluate the protein expression levels of TRIM2 and BNIP3 (*n* = 8 biologically independent samples). The quantification of the BNIP3 western blot bands at different time points is presented on the right. β-actin was used as a loading control. **G** The protein changes of BNIP3 in three groups of Caco-2 cells (transfected with HA-Vector plasmid and treated with DMSO,

transfected with HA-TRIM2 plasmid and treated with DMSO, transfected with HA-TRIM2 plasmid and treated with MG132) after CHX treatment for indicated times. The western blot detected the half-life of BNIP3 in three groups, with β-actin serving as a loading control (*n* = 6 biologically independent samples). **H** The protein alterations of BNIP3 in three groups (treated with si-Control, treated with si-TRIM2, treated with si-TRIM2 and HA-TRIM2 plasmid) of Caco-2 cells following CHX treatment for the specified times. The half-life diagram of BNIP3 in the three groups is presented on the right. β-actin was used as a loading control (*n* = 9 biologically independent samples). **I** The protein changes of BNIP3 in three groups (co-transfected with Flag-BNIP3 and HA-Vector, co-transfected with Flag-BNIP3 and HA-TRIM2, and co-transfected with Flag-BNIP3-K130R and HA-TRIM2) of Caco-2 cells after CHX treatment for indicated times. Western blot detected the half-life of Flag-BNIP3 in the three groups (*n* = 8 biologically independent samples). β-actin was used as a loading control. **J** Caco-2 cells were treated with si-BNIP3, and immunofluorescence was performed to detect the co-localization and distribution of TRIM2 and Flag-BNIP3 in the different groups (si-BNIP3+Flag-BNIP3-WT + DMSO, the experimental groups were as follows: si-BNIP3+Flag-BNIP3-WT + MG132, si-BNIP3+Flag-BNIP3-K130R + DMSO, and si-BNIP3+Flag-BNIP3-K130R + MG132. Scale bars are 10 μm. All results are expressed as the mean ± SD. Statistical significance was determined using one-way ANOVA followed by Tukey's test, with the following levels of significance: *$p < 0.05$, **$p < 0.01$, ***$p < 0.001$, ****$p < 0.001$.

plates at a density of $1 \times 10^4$ cells per well and cultured under standard conditions for 24 hours to ensure adherence. After H/R treatment, 10 μL of the CCK-8 reagent was added to each well, followed by a 2-hour incubation at 37 °C. During this period, the water-soluble tetrazolium salt present in the CCK-8 reagent is reduced by viable cells, resulting in the formation of an orange-colored formazan product. Absorbance was measured at 450 nm using a Varioskan LUX microplate reader (ThermoFisher Scientific, Waltham, MA, USA), with values directly proportional to viable cell numbers. Cell viability was expressed as a percentage relative to untreated controls.

### Lactate dehydrogenase (LDH) assay
The release of LDH into the cell culture supernatant is a rapid and crucial event that occurs when the plasma membrane is damaged[21]. This phenomenon is a hallmark of cells undergoing apoptosis, necrosis, and other forms of cellular damage[21]. The LDH activity in the supernatants of cell cultures was quantified using an LDH assay kit (A020-2-2, JianChan Bioengineering Institute, Nanjing, China). IEC-6 or Caco-2 cells were seeded in 6-well plates and cultured to 70–80% confluence prior to H/R treatment. Following hypoxia/reoxygenation (H/R) treatment, the culture supernatants were collected for LDH analysis (representing LDH release). To determine total cellular LDH content (used for normalization), the remaining adherent cells were lysed by incubation with 0.2% Triton X-100 for 15 minutes at 37 °C. The resulting cell lysate supernatant was then collected. LDH activity in both the conditioned culture supernatants (released LDH) and the cell lysate supernatants (total LDH) was assayed according to the manufacturer's instructions. Briefly, samples were incubated with the LDH assay reagent containing NADH and substrate solutions at 37 °C for 30 minutes. LDH enzymatic activity catalyzed the conversion of substrates, and the reaction kinetics were measured via absorbance at 450 nm using a Varioskan LUX microplate reader (ThermoFisher Scientific, Waltham, MA, USA). LDH release was calculated as follows: LDH Release (%) = [(LDH Activity in Conditioned Supernatant) × (Volume of Supernatant Collected)] / [(LDH Activity in Conditioned Supernatant × Volume of Supernatant) + (LDH Activity in Lysate Supernatant × Volume of Lysate)] × 100%. This represents the percentage of total cellular LDH activity released into the culture medium.

### Pathological tissue staining and immunohistochemistry
Intestine tissues from mice were fixed in 4% paraformaldehyde, embedded in paraffin, and sectioned into 4-μm-thick slices. Hematoxylin and eosin (H&E) staining was performed using commercial kits (Solarbio, Beijing, China) following the manufacturer's protocol. The extent of intestinal injury

was quantified according to the criteria of *Chiu*'s score[58]. The immunohistochemical staining (IHC) was performed in accordance with the previous research[31]. The specific antibodies employed in this study included anti-TRIM2 (Proteintech, China; 20356-1-AP, 1:100), anti-Bax (Proteintech, China; 50599-2-Ig, 1:400), anti-BNIP3 (Cell Signaling Technology, America; 44060S, 1:100), anti-Bcl2 (Proteintech, China; 12789-1-AP, 1:200), and anti-cleaved Casp3 (Proteintech, China; 19677-1-AP, 1:100). Images were observed and captured using a Sunny Optical Technology HS6 microscope with ImageScope software. Quantification was performed using ImageJ (National Institutes of Health) on six randomly selected fields per sample. All evaluations were conducted by investigators blinded to experimental groups.

### qRT-PCR
Total RNA was extracted from intestine tissues and intestinal epithelial cells and reverse-transcribed into cDNA using TRIzol reagent (T9108, Takara) and a cDNA synthesis kit (11141ES60, YEASEN, Shanghai, China), following the instructions provide by the manufacturers. Quantitative RT-PCR was conducted using SYBR Green PCR Master Mix (11201ES08, YEASEN, Shanghai, China) and the CFX Connect system (Bio-Rad, CA, USA). mRNA expression was normalized to β-actin expression and the values were quantitatively analyzed using the $2^{-\Delta\Delta Ct}$ method. The primer pairs were synthesized by Tsingke and are listed in Supplementary Data 1.

### Western blot
Total protein was extracted from mouse intestine tissues or cultured cells using RIPA lysis buffer (WB3100, NCM Biotech, Suzhou, China) containing 1% protease inhibitor cocktail (P001, NCM Biotech, Suzhou, China) on ice. Lysates were centrifuged at 12,000 rpm at 4 °C for 15 minutes, and protein concentrations were determined using the BCA Protein Assay Kit (WB6501, NCM Biotech, Suzhou, China). Equal amounts of protein were mixed with 2× Laemmli loading buffer (2.1% SDS, 0.01% bromophenol blue, 26.3% glycerol, 6.58% Tri-HCl pH 6.8 and 0.5% β-mercaptoethanol), denatured at 95 °C for 5 min, and resolved on 10% SDS-PAGE gels (20325ES62, YEASEN, Shanghai, China). Proteins were transferred to nitrocellulose membranes (PALL Corporation, Port Washington, NY, USA) and blocked with 5% non-fat milk in PBS-Tween 30 (PBST) for 1 h at room temperature. Membranes were incubated overnight with the primary antibody at 4 °C, washed, and probed with HRP-conjugated secondary antibody (31460 and 31430, Thermo Scientific, USA) for 2 h at room temperature. The bands of target proteins were visualized with an ECL chemiluminescent substrate kit (SB-WB004, ShareBio, Shanghai, China),

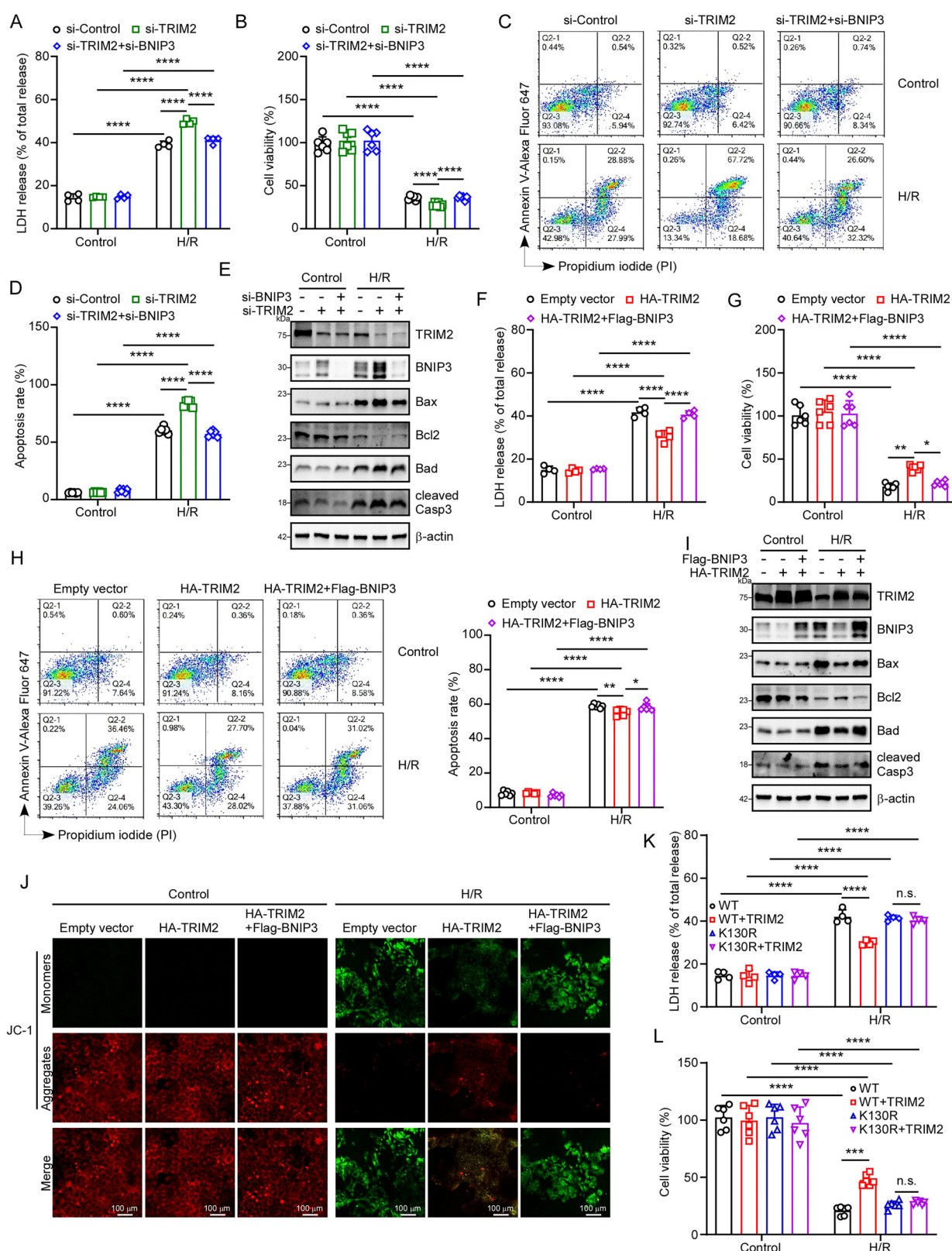

and protein levels were quantified using ImageJ software (National Institutes of Health) normalized with β-actin.

## Co-IP and Ubiquitination assay

To detect the interaction between proteins, HEK293T cells were cultured and co-transfected with the indicated plasmids using Polyethylenimine

Linear (PEI) MW40000 (YEASEN, Shanghai, China). Following a 48-hours transfection period, HEK293T cells were treated with 10 μM proteasome inhibitor MG132 for 4 hours before lysis. The lysis buffer consists of 0.5% lubrol-px, 50 mM KCl, 2 mM CaCl₂, 20% glycerol, 50 mM Tris-HCl, 3 mM sodium vanadate, and 2 mM NaF. The specific antibodies (1 μg) were used to immunoprecipitate the whole-cell lysates mixed with pre-activated

**Fig. 8 | TRIM2 targets BNIP3 to inhibit II/R-induced apoptosis. A** The LDH activity of Caco-2 cell supernatants was evaluated in six experimental groups ($n = 4$ biologically independent samples): Control + si-Control, Control + si-TRIM2, Control + si-TRIM2 + si-BNIP3, H/R + si-Control, H/R + si-TRIM2, and H/R + si-TRIM2 + si-BNIP3. **B** The cell viability of Caco2 was assessed using the CCK-8 assay in the six experimental groups ($n = 6$ biologically independent samples). **C, D** Flow cytometry was employed to ascertain the extent of apoptosis in Caco-2 cells subjected to disparate treatments ($n = 6$ biologically independent samples). **E** Western blot analysis was employed to ascertain the protein expression levels of TRIM2, BNIP3, BAX, Bcl2, BAD and cleaved CASP3 in Caco-2 cells subjected to TRIM2 and BNIP3 protein knockdown and H/R treatment. **F** The LDH activity of Caco-2 cells in the aforementioned six groups was also evaluated ($n = 4$ biologically independent samples). The supernatants of the two cell lines were analyzed in six groups (Control+Empty vector, Control+HA-TRIM2, Control+HA-TRIM2+-Flag-BNIP3, H/R+Empty vector, H/R + HA-TRIM2, H/R + HA-TRIM2+Flag-BNIP3). **G** The cell viability of Caco-2 was determined through the use of a CCK8 assay, which was conducted on six experimental groups ($n = 6$ biologically

independent samples). **H** Flow cytometry was employed to ascertain the extent of apoptosis in Caco-2 cells subjected to disparate treatments ($n = 8$ biologically independent samples). **I** Western blot was employed to ascertain the protein expression of TRIM2, BNIP3, BAX, Bcl2, BAD and cleaved CASP3 in Caco-2 cells subjected to TRIM2 and BNIP3 protein overexpression and H/R treatment. **J** Following transfection of the HA-TRIM2 and Flag-BNIP3 plasmids and subsequent treatment with H/R, Caco-2 cells were stained with JC-1 for 20 minutes at 37 °C. Subsequently, the cells were observed using a laser scanning confocal microscope. Scale bars are 100 μm (Caco-2). **K** LDH activity was measured in the supernatants of Caco-2 cells in eight groups ($n = 4$ biologically independent samples): Control+Flag-BNIP3-WT + HA-Vector, Control+Flag-BNIP3-WT + HA-TRIM2, Control+Flag-BNIP3-K130R + HA-Vector, Control+Flag-BNIP3-K130R + HA-TRIM2. **L** The cell viability of Caco-2 was assessed using the CCK8 assay in eight groups ($n = 6$ biologically independent samples). All results are expressed as the mean ± SD. Statistical significance was determined using one-way ANOVA followed by Tukey's test, with *$p < 0.05$, ** $p < 0.01$, ***$p < 0.001$, and ****$p < 0.0001$ indicating statistical significance.

Protein A/G Agarose (Beyotime, China) at 4 °C overnight. Subsequently, the co-immunoprecipitation products were obtained by washing the samples four times and analyzed by Western blot after being mixed with 2×loading buffer. For the ubiquitination assay, GFP-TRIM2, HA-Ub (WT) or its mutants (HA-Ub-K6O, HA-Ub-K11O, HA-Ub-K27O, HA-Ub-K29O, HA-Ub-K33O, HA-Ub-K48O, HA-Ub-K63O, HA-Ub-K6R, HA-Ub-K11R, HA-Ub-K27R, HA-Ub-K29R, HA-Ub-K33R, HA-Ub-K48R, and HA-Ub-K63R), and Flag-BNIP3 or its mutants (Flag-K45-BNIP3, Flag-K60-BNIP3, Flag-K90-BNIP3, Flag-K104-BNIP3, Flag-K111-BNIP3, Flag-K112-BNIP3, Flag-K130-BNIP3, Flag-K135-BNIP3, Flag-K138-BNIP3, Flag-K152-BNIP3, Flag-K153-BNIP3 and Flag-K163-BNIP3) were co-transfected into HEK293T cells. To investigate the ubiquitination of Flag-BNIP3, the lysates were subjected to immunoprecipitation using an anti-Flag tag antibody (1 μg) at 4 °C overnight, and immunoblotting analysis was performed using antibodies specific for HA, K48-Ub, and K63-Ub.

### GST pull-down assay
To investigate the direct interaction between TRIM2 and BNIP3, recombinant GST-TRIM2 protein and control GST proteins were expressed and purified in *Escherichia coli* BL21. The GST pull-down assay was performed as previously described[59]. In brief, following washing and activation with glutathione Sepharose 4B (GE Healthcare, Sweden), lysates from HEK293T cells transfected with Flag-BNIP3 or its deletion mutants were incubated with recombinant GST or GST-TRIM2 proteins at 4 °C overnight. Subsequently, the complexes were centrifuged and rinsed thoroughly, and then detected by Western blot with anti-Flag antibody.

### Cellular immunofluorescence
Following treatment with H/R, Caco-2 or IEC-6 cells cultured in 24-well plates ($1 \times 10^4$ cells/well) were fixed with freshly prepared 4% paraformaldehyde for 30 minutes. Subsequently, the cells were permeabilized in 0.5% Triton X-100 for 20 minutes and blocked with normal goat serum (AR0009, BOSTER, Wuhan, China) for one hour at room temperature. Subsequently, the cells were incubated with anti-TRIM2 (Proteintech, China; 67342-1-Ig, 1:200) and anti-BNIP3 (Cell Signaling Technology, USA; 44060S, 1:400) antibodies for a period of 24 hours at 4 °C. After washing with PBS, cells were stained for 2 h at room temperature in the dark with fluorescent secondary antibodies (Solarbio: SF134 [Alexa Fluor 488], K1031G-Cy3 [Cy3]) and counterstained with 4′,6-diamidino-2-phenylindole (DAPI; Solarbio, S2110) to label nuclei. Images were acquired using a Leica TCS SP8 laser scanning confocal microscope (Leica Microsystems, Wetzlar, Germany).

### Cell apoptosis assay
IEC-6 and Caco-2 cells were cultured in 6-well plates ($1 \times 10^6$ cells/well) for a period of 24 hours at 37 °C and 5% CO₂. The apoptotic level of cells subjected to different treatments was assessed using the Annexin

V-Alexa Fluor 647/PI Apoptosis Detection Kit (40304ES60, YEASEN, Shanghai, China), in accordance with the manufacturer's instructions. The total apoptotic rate, comprising early-stage and late-stage apoptosis, was analyzed using a Beckman flow cytometer with FlowJo v10 software (FlowJo; BD Biosciences). Intestinal tissues were embedded in paraffin and sectioned at a thickness of 4 μm. The TMR (red) Tunnel Cell Apoptosis Detection Kit (G1502, Servicebio, Wuhan, China) was used to fix and label the sections, in accordance with the manufacturer's instructions. The TUNEL-positive cells were observed and photographed under a confocal laser scanning microscope (Leica, Germany), and the ratio of TUNEL-positive cells to DAPI-positive nuclei was calculated using Image J software.

### JC-1 staining
The mitochondrial membrane potential (MMP) was determined using the Mitochondrial Membrane Potential Assay Kit with JC-1 (Beyotime, China). IEC-6 or Caco-2 cells, cultured in 24-well plates ($5 \times 10^4$ cells/well) in different groups, were incubated with JC-1 staining working solution (50 μl JC-1 200 × in 8 mL ddH₂O) after rinsing with PBS buffer at a CO₂ incubator for 20 minutes. Subsequently, the cells were washed twice with pre-cooled 1 × JC-1 staining buffer. Images were obtained using a laser confocal microscope (Leica, Germany), and the ratios of red and green fluorescence intensity were calculated using Image J software.

### Statistics and Reproducibility
The statistical analysis was conducted using GraphPad Prism version 10 (GraphPad Software, Inc.), and all data were expressed as mean ± standard deviation (SD). The two-tailed Student's *t*-test was employed for the statistical analysis of data from two groups with normal distributions. Additionally, comparisons among multiple groups were evaluated using one-way analysis of variance (ANOVA) with Bonferroni correction. For grouped analyses, either multiple unpaired t-tests or two-way ANOVA with Tukey's multiple comparisons test were performed. In the event that the data did not meet the criteria for normality, nonparametric rank-sum tests were employed for comparison. $p < 0.05$ was considered statistically significant. Each experiment was performed with at least three to nine biological replicates. For information on sample sizes and tests applied, see the figure legends.

### Reporting summary
Further information on research design is available in the Nature Portfolio Reporting Summary linked to this article.

### Data availability
The data analyzed during this study are included in this published article. The source data behind the graphs in the paper can be found in Supplementary Data 2. Further information and requests for resources and

reagents should be directed to and will be fulfilled by the lead contact, Yong Li (liyong@ncu.edu.cn).

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

## Acknowledgements

We appreciate Dr. Hongbin Shu and Dr. Shu Li (School of Medicine, Wuhan University) for providing HA-Ub mutant plasmids. We thank Alan Jiang (The First Affiliated Hospital, Jiangxi Medical College, Nanchang University) for microscopy assistance. This work was supported by grants from the National Natural Science Foundation of China (32560166, 32170793 and 82260114), Jiangxi Provincial Natural Science Foundation (20224BAB206007 and 20224ACB216013).

## Author contributions

Y.L. and X.H. conceived and designed the study, prepared the figures, analyzed data, and participated in the paper writing. Y.L. and X.Z. supervised the study and planned experiments. J.N. and C.M. performed the experiments, analyzed the data, and wrote the first draft of the manuscript. A.W., Y.W., Y.H., M.J., T.C., and J.T. conducted experiments during the study. A.W. and H.C. bred and genotyped the mice. Y.W. and C.F. critically revised the manuscript for important intellectual content. YL and XH revised the manuscript. All authors approved and contributed to the final version of the manuscript.

## Competing interests

The authors declare no competing interests.
