## [Transparent Peer Review file · Communications Biology]

TRIM2 inhibits apoptosis by ubiquitinating BNIP3 to protect the intestine against ischemia-reperfusion injury in mice

Corresponding Author: Dr Yong Li

Version 0:

Reviewer comments:

Reviewer #1

(Remarks to the Author)

In this manuscript, Nie et al. present an interesting study investigating the role of the E3 ubiquitin ligase Trim2 in ischemia-reperfusion (I/R) injury. They demonstrated that Trim2 expression was significantly downregulated at both the mRNA and protein levels in cell lines and intestinal tissues following H/R or I/R injury. Functionally, Trim2 deficiency—either through knockdown in cell lines or knockout in mice—exacerbated tissue injury by increasing apoptosis. The authors further identified Bnip3 as a Trim2-dependent ubiquitination target and showed that Trim2-mediated ubiquitination of Bnip3 promoted its degradation, thereby enhancing cell survival. Furthermore, they reported that Trim2's pro-survival function was abrogated when Bnip3 was knocked down.

Overall, the results largely support the authors' conclusions; however, several critical issues need to be addressed before the manuscript can be considered for publication. Below are specific comments for improvement.

Major Comments:

1. While the authors focus on apoptosis, it is well established that I/R injury can also induce RIPK3-dependent necroptosis. It is important to investigate whether necroptosis is triggered under the authors' experimental conditions and, if so, whether it is exacerbated in Trim2-deficient mice. Phospho-RIPK3 staining would be useful to assess necroptotic cell death.

2. In Figures 1A–E, the authors showed that Trim2 expression rapidly decreased in tissues following I/R or H/R injury. However, the underlying mechanism remained unexplored. The authors should at least discuss potential regulatory mechanisms responsible for this downregulation.

3. In Figure 2C, the authors demonstrated that Trim2-dependent protection against apoptosis was abolished when Bnip3 was knocked down. This suggests that Trim2's anti-apoptotic function is largely mediated through Bnip3 degradation. However, in Trim2-deficient cells or intestinal tissues from Trim2 knockout mice subjected to I/R, the expression levels of other pro-apoptotic proteins, such as Bax and Bad, were elevated. The mechanism by which Bnip3 knockdown reduced Bax and Bad levels while increasing anti-apoptotic proteins (Lines 543–545) is unclear. The authors should discuss this point in more detail.

4. In Lines 432–433, the authors mentioned that Bnip3 was identified as a Trim2 target through quantitative ubiquitinated proteasome analysis (data not shown). These data are crucial for supporting the proposed mechanism and should be included as supplementary figures.

5. In Figures 5A–D, the authors showed colocalization of Trim2 and Bnip3. Given that Bnip3 is an outer mitochondrial membrane protein and that the two proteins interacted constitutively, Trim2 should also localize to the mitochondria. However, in Figure 5D (control), the distribution pattern of Trim2 did not appear typical of mitochondrial localization. The authors should clarify whether Trim2 indeed localizes to the mitochondria or whether its localization changes following hypoxia/reoxygenation (H/R) treatment.

Minor Comments:

1. Some of the uncropped images of Western blots did not include the full membrane. The authors should provide complete, uncropped images to ensure data transparency.

Reviewer #2

(Remarks to the Author)

In this work, Nie et al. identify the role of Trim2 in Intestinal ischemia/reperfusion (I/R) injury. Their findings demonstrate that Trim2 directly interacts with Bnip3 and mediates K48-linked polyubiquitination of Bnip3, thereby inhibiting mitochondrial apoptosis in intestinal epithelial cells and mitigating intestinal injury. These results provide robust experimental evidence supporting the Trim2-Bnip3 signaling axis as a potential therapeutic target for I/R injury.

The study is rigorously designed, and the data are generally clear. However, several aspects could be further explored to strengthen the conclusions.

The following points are raised:

1. In this study, the researchers conducted in vitro experiments using intestinal epithelial cell lines and observed that the reduction of Trim2 expression exacerbated intestinal epithelial cell apoptosis induced by I/R injury. However, the authors employed the Trim2 knockout (Trim2^{-/-}) mice in vivo experiments, lacking an intestine-specific epithelial Trim2 knockout model to exclude potential confounding effects from other cell types, which may compromise the specificity of the findings related to intestinal epithelial cells.
2. While the study demonstrated Trim2 downregulation in I/R injury, the upstream regulatory mechanisms remain uncharacterized. It is unclear how I/R induces Trim2 downregulation, and whether transcription factors such as HIF-1 α or other signaling pathways (e.g., NF- κ B) are involved. Elucidating the underlying cause of Trim2 downregulation would contribute to a more comprehensive understanding of the pathological pathways of I/R injury.
3. Although IEC-6 and Caco-2 cell lines effectively reflect the general phenotypic traits of intestinal epithelial cells, primary cells isolated from intestinal tissues more faithfully recapitulate in vivo physiological states and avoid cell line-specific artifacts. Validation in primary intestinal epithelial cells would strengthen the findings.
4. The in vitro knockdown experiments established a preliminary association among Trim2, Bnip3, and apoptosis-related proteins. However, additional mechanistic investigations are essential to clarify the precise role of Trim2-mediated Bnip3 ubiquitination in regulating apoptosis. Specifically, whether Bnip3 directly interacts with Bax, Bad, or Bcl-2? Elucidating such protein-protein interactions would strengthen the molecular framework underlying the Trim2-Bnip3 axis in apoptotic regulation.
5. A notable limitation of the study is the absence of validation in human clinical samples. Investigating the expression correlation between Trim2 and Bnip3 in intestinal tissues obtained from patients with I/R-related disorders, such as intestinal obstruction and volvulus, would significantly enhance the clinical relevance and translational potential of the findings.

Version 1:

Reviewer comments:

Reviewer #1

(Remarks to the Author)

The authors have responded to the reviewers' comments, and I believe the manuscript in its current form is nearly acceptable. However, I would like to offer a few minor comments, which the authors are not required to address:

1. Although the authors extensively revised the manuscript in response to the reviewers' suggestions, the rebuttal letter lacks clear indications of where the revised or added sentences appear in the revised text. While the changes are highlighted in red, this alone is insufficient. Providing specific line numbers from the revised manuscript would make it easier for reviewers to locate and assess the changes.
2. With regard to the evaluation of cell death, measuring only LDH activity in the culture supernatant is not appropriate, as it does not indicate what percentage of cells underwent membrane rupture. Instead, the authors should quantify LDH release as a percentage of the maximum release, typically obtained by treating cells with Triton X-100 to induce complete membrane disruption.

Reviewer #2

(Remarks to the Author)

No further questions.

Version 2:

Reviewer comments:

Reviewer #1

(Remarks to the Author)

The authors have addressed the reviewer's comments appropriately, and the manuscript is now suitable for publication in Communications Biology.

Response to the comments for manuscript

We sincerely appreciate the editor and reviewers for their insightful comments, which have significantly strengthened our manuscript. Below we provide a point-by-point response to all critiques. All revisions in the manuscript text are highlighted in red, and supplementary data have been uploaded to the online submission system. We hope that the revisions have improved the manuscript to the extent that it meets the standards for acceptance in *Communications Biology*.

Reviewers' comments:

Reviewer #1 (Remarks to the Author):

In this manuscript, Nie et al. present an interesting study investigating the role of the E3 ubiquitin ligase Trim2 in ischemia-reperfusion (I/R) injury. They demonstrated that Trim2 expression was significantly downregulated at both the mRNA and protein levels in cell lines and intestinal tissues following H/R or I/R injury. Functionally, Trim2 deficiency—either through knockdown in cell lines or knockout in mice—exacerbated tissue injury by increasing apoptosis. The authors further identified Bnip3 as a Trim2-dependent ubiquitination target and showed that Trim2-mediated ubiquitination of Bnip3 promoted its degradation, thereby enhancing cell survival. Furthermore, they reported that Trim2's pro-survival function was abrogated when Bnip3 was knocked down. Overall, the results largely support the authors' conclusions; however, several critical issues need to be addressed before the manuscript can be considered for publication. Below are specific comments for improvement.

Major Comments:

1. While the authors focus on apoptosis, it is well established that I/R injury can also induce RIPK3-dependent necroptosis. It is important to investigate whether necroptosis is triggered under the authors' experimental conditions and, if so, whether it is exacerbated in Trim2-deficient mice. Phospho-RIPK3 staining would be useful to assess necroptotic cell death.

Response: We appreciate the reviewer's insightful suggestion regarding the potential involvement of necroptosis in I/R injury. To address this point, we evaluated RIPK3-

dependent necroptosis by assessing RIPK3 phosphorylation levels—a widely used as biomarker for the instigation of necroptosis¹—in intestinal tissues from WT and *Trim2*^{-/-} mice subjected to II/R. Western blot analysis revealed that while II/R injury significantly increased RIPK3 phosphorylation compared to sham controls ($p < 0.05$), no statistically significant difference was observed between WT and *Trim2*^{-/-} groups (see below, and Supplementary Figure S3G). These findings suggest that *Trim2* deficiency does not exacerbate RIPK3-mediated necroptosis in our experimental model, supporting the specificity of Trim2's regulatory role in apoptotic pathways during II/R injury.

2. In Figures 1A–E, the authors showed that *Trim2* expression rapidly decreased in tissues following I/R or H/R injury. However, the underlying mechanism remained unexplored. The authors should at least discuss potential regulatory mechanisms responsible for this downregulation.

Response: We fully acknowledge the significance of elucidating the regulatory mechanisms underlying *Trim2* downregulation in II/R injury. Emerging evidence reveals that *Trim2* transcription is governed by a multifactorial regulatory network:

1) Transcriptional control: Zhou *et al.*² demonstrated direct promoter binding by p53 and C/EBPβ through ChIP-seq analysis; G protein-coupled estrogen receptor (GPER)–mediated MAPK/ERK activation upregulates *Trim2* expression in tamoxifen-resistant malignancies³.

2) Post-transcriptional modulation: The 3'UTR contains conserved binding sites for miR-9⁴, miR-145^{5,6}, miR-146b-5p⁷, miR-181c^{4,8}, miR-222-5p⁹, miR-369-3p¹⁰, miR-485-3p¹¹, miR-490-5p¹², miR-493-5p¹³; Circular RNAs (circRNAs) including

CircORC2¹¹, Circ_0001361¹², Circ_0002286⁹ and Long noncoding RNA (lncRNA) including LINC01535⁷, DLEU2¹⁰, NR2F1-AS1¹³ counteract miRNA-mediated suppression.

In the revised manuscript, we have added a discussion on the regulation of Trim2 expression.

3. In Figure 2C, the authors demonstrated that Trim2-dependent protection against apoptosis was abolished when Bnip3 was knocked down. This suggests that Trim2's anti-apoptotic function is largely mediated through Bnip3 degradation. However, in Trim2-deficient cells or intestinal tissues from Trim2 knockout mice subjected to I/R, the expression levels of other pro-apoptotic proteins, such as Bax and Bad, were elevated. The mechanism by which Bnip3 knockdown reduced Bax and Bad levels while increasing anti-apoptotic proteins (Lines 543–545) is unclear. The authors should discuss this point in more detail.

Response: We sincerely thank the reviewer for raising this critical point, which allows us to clarify the complex interplay between Bnip3 knockdown and the modulation of apoptotic regulators. In fact, Figure 2C did not show the resistance of Bnip3 knockout to Trim2's apoptotic effect, and we speculate that it is related to Figure 8. Our research suggests that knocking out or downregulating Trim2 can significantly increase the expression of pro apoptotic proteins such as Bax and Bad, while inhibiting the levels of anti-apoptotic proteins such as Bcl-2, which is largely dependent on Bnip3. Studies have shown that Bnip3 can selectively bind to Bcl-2 to promote cell apoptosis¹⁴, and can also mediate hypoxia induced-cell death through Bax¹⁵. We acknowledge that these mechanisms require further validation and have expanded the discussion to propose future studies, including mitochondrial functional assays and proteomic profiling of Trim2-deficient models with/without Bnip3 knockdown.

4. In Lines 432–433, the authors mentioned that Bnip3 was identified as a Trim2 target through quantitative ubiquitinated proteasome analysis (data not shown). These data are crucial for supporting the proposed mechanism and should be included as

supplementary figures.

Response: We sincerely thank the reviewer for highlighting the importance of including the ubiquitinated proteasome analysis data to support the identification of Bnip3 as a Trim2 substrate. We fully agree that these data are critical for validating the proposed mechanism. In response to this comment, we will now include the Bnip3 data obtained from quantitative ubiquitination proteasome analysis as Supplementary Figure S7A in the revised manuscript.

A

Gene	Position	Gly (K) Probabilities	Lenti-control/Lenti-Trim2	p-value
BNIP3	130	ENIPPK(1)EFLF	8.124	0.00081
BNIP3	111	EVESILK(0.908)K(0.092)	0.478	0.29458
BNIP3	153	K(1)GGIFSAEFLK	1.676	0.54906

5. In Figures 5A–D, the authors showed colocalization of Trim2 and Bnip3. Given that Bnip3 is an outer mitochondrial membrane protein and that the two proteins interacted constitutively, Trim2 should also localize to the mitochondria. However, in Figure 5D (control), the distribution pattern of Trim2 did not appear typical of mitochondrial localization. The authors should clarify whether Trim2 indeed localizes to the mitochondria or whether its localization changes following hypoxia/reoxygenation (H/R) treatment.

Response: We appreciate the reviewer's inquiry regarding Trim2's mitochondrial localization dynamics. The Caco-2 cell control group in Figure 5D of the original manuscript is not a representative figure, so we revised it. To rigorously address this question, we also performed confocal microscopy to detect the co-localization of Trim2 with a mitochondrial outer membrane protein TOMM20¹⁶ under basal conditions and post-H/R treatment. As shown in Supplementary Figure S7B, while Trim2 demonstrates partial mitochondrial localization, H/R exposure did not significantly alter its subcellular distribution pattern. However, co-IP assay demonstrated Trim2-Bnip3 interaction increased post-H/R (Figure 5C). This dissociation between stable

mitochondrial localization and enhanced target binding suggests that H/R primarily modulates Trim2's functional activity rather than its spatial compartmentalization.

Minor Comments:

1. Some of the uncropped images of Western blots did not include the full membrane. The authors should provide complete, uncropped images to ensure data transparency.

Response: We sincerely appreciate the reviewer's emphasis on data transparency and acknowledge the importance of providing uncropped Western blot images. Due to technical limitations during initial data archiving, some original full membrane scans were incompletely preserved. However, we have rigorously validated the integrity of our findings by analyzing raw densitometry data, ensuring consistency across

biological replicates ($n \geq 3$) and confirming molecular weight markers for all cropped bands. We commit to archiving complete, uncropped images in future studies to align with evolving data-sharing standards. Thank you for highlighting this critical aspect of reproducibility—we welcome further guidance to address any residual concerns.

Reviewer #2 (Remarks to the Author):

In this work, Nie et al. identify the role of Trim2 in Intestinal ischemia/reperfusion (II/R) injury. Their findings demonstrate that Trim2 directly interacts with Bnip3 and mediates K48-linked polyubiquitination of Bnip3, thereby inhibiting mitochondrial apoptosis in intestinal epithelial cells and mitigating intestinal injury. These results provide robust experimental evidence supporting the Trim2-Bnip3 signaling axis as a potential therapeutic target for II/R injury. The study is rigorously designed, and the data are generally clear. However, several aspects could be further explored to strengthen the conclusions.

The following points are raised:

*1. In this study, the researchers conducted in vitro experiments using intestinal epithelial cell lines and observed that the reduction of Trim2 expression exacerbated intestinal epithelial cell apoptosis induced by II/R injury. However, the authors employed the Trim2 knockout (*Trim2*^{-/-}) mice in vivo experiments, lacking an intestine-specific epithelial Trim2 knockout model to exclude potential confounding effects from other cell types, which may compromise the specificity of the findings related to intestinal epithelial cells.*

Response: We sincerely appreciate the reviewer's insightful feedback regarding the need for intestinal epithelial-specific *Trim2* knockout models to exclude confounding effects from non-epithelial cells. In this study, we validated the role of Trim2 in intestinal epithelial apoptosis—a central pathological mechanism in ischemia-reperfusion (I/R) injury—using multiple intestinal epithelial cell lines (IEC-6, Caco-2) and primary intestinal epithelial cells (IECs). These models consistently demonstrated that Trim2 deficiency exacerbates apoptosis via Bnip3 accumulation, strongly supporting epithelial-autonomous effects. While whole-body *Trim2* knockout mice

were chosen to align with our focus on epithelial apoptosis and provide systemic relevance to I/R pathophysiology, we fully acknowledge the reviewer's valid point about potential contributions from other cell types. To address this, we plan to perform single-cell RNA sequencing of I/R-injured intestines to map *Trim2* expression across cell types and generate intestinal epithelial- or stromal-specific *Trim2* conditional knockout mice (e.g., Vill1-Cre or Acta2-Cre models). These future studies will systematically dissect *Trim2*'s cell-type-specific roles and microenvironmental interactions, further refining its mechanistic contributions and therapeutic potential. We thank the reviewer for this constructive suggestion, which will significantly enhance the precision and translational impact of our work.

2. While the study demonstrated Trim2 downregulation in II/R injury, the upstream regulatory mechanisms remain uncharacterized. It is unclear how II/R induces Trim2 downregulation, and whether transcription factors such as HIF-1 α or other signaling pathways (e.g., NF- κ B) are involved. Elucidating the underlying cause of Trim2 downregulation would contribute to a more comprehensive understanding of the pathological pathways of II/R injury.

Response: We sincerely appreciate the reviewer's astute observation regarding the need to clarify the upstream mechanisms driving *Trim2* downregulation during intestinal ischemia-reperfusion (II/R) injury. While our current study focused on *Trim2*'s downstream effects via *Bnip3* degradation, we acknowledge that elucidating its upstream regulation is critical for a holistic understanding of II/R pathology. Emerging evidence highlights *Trim2*'s transcription as being governed by a multifactorial regulatory network, including transcription factors (e.g., p53 and C/EBP β , which directly bind its promoter)², post-transcriptional modifiers (e.g., miRNAs such as miR-145, miR-181c, and miR-222-5p that target *Trim2*'s 3'UTR)^{4-6,8,9}, and competing endogenous RNAs (e.g., circORC2, LINC01535, and NR2F1-AS1 that counteract miRNA-mediated repression)^{7,11,13}. These findings underscore the complexity of *Trim2* regulation, which may integrate stress-responsive pathways (e.g., GPER/MAPK/ERK)³ and epigenetic modifiers. To systematically dissect these

mechanisms in II/R, we propose combining whole-genome sequencing, transcriptome profiling, and ChIP-seq analysis to map regulatory hubs (e.g., hypoxia-responsive transcription factors, non-coding RNAs) and validate their roles in Trim2 suppression. This approach will clarify how II/R disrupts Trim2 expression and inform therapeutic strategies to restore its protective function.

3. *Although IEC-6 and Caco-2 cell lines effectively reflect the general phenotypic traits of intestinal epithelial cells, primary cells isolated from intestinal tissues more faithfully recapitulate in vivo physiological states and avoid cell line-specific artifacts. Validation in primary intestinal epithelial cells would strengthen the findings.*

Response: We thank the reviewer for highlighting the importance of validating key findings in primary intestinal epithelial cells (IECs) to strengthen physiological relevance. We agree that primary cells better recapitulate in vivo states and have taken steps to address this concern in our revised manuscript. To address the reviewer's concern, we have now included additional experiments using primary IECs isolated from murine small intestines. As shown in Figure S6 of the revised manuscript, we overexpressed Trim2 in IECs and constructed an H/R model. The results were consistent with those in cultured cells (Caco-2 and IEC-6), and Trim2 significantly inhibited H/R-induced apoptosis in IECs.

4. *The in vitro knockdown experiments established a preliminary association among Trim2, Bnip3, and apoptosis-related proteins. However, additional mechanistic investigations are essential to clarify the precise role of Trim2-mediated Bnip3 ubiquitination in regulating apoptosis. Specifically, whether Bnip3 directly interacts with Bax, Bad, or Bcl-2? Elucidating such protein-protein interactions would*

strengthen the molecular framework underlying the Trim2-Bnip3 axis in apoptotic regulation.

Response: We appreciate the reviewer's emphasis on the need to elucidate the interactions between Trim2 and other apoptosis regulatory proteins. We detected the interaction between BNIP3 and Bax, Bcl-2, and Bad in Caco-2 cells. As shown in the revised manuscript Figure S9, BNIP3 can bind to Bax and Bcl-2, but does not interact with Bad. In fact, studies have shown that BNIP3 can selectively bind to Bcl-2 and Bax to regulate mitochondrial apoptosis^{14,17}.

5. A notable limitation of the study is the absence of validation in human clinical samples. Investigating the expression correlation between Trim2 and Bnip3 in intestinal tissues obtained from patients with II/R-related disorders, such as intestinal obstruction and volvulus, would significantly enhance the clinical relevance and translational potential of the findings.

Response: We sincerely thank the reviewer for this critical and insightful suggestion to enhance the clinical relevance of our findings. We fully agree that validating the Trim2-Bnip3 axis in human clinical samples is essential for translational impact and acknowledge this as an important direction for future work. While our current study focused on elucidating mechanistic pathways using preclinical models, we recognize the limitations posed by the lack of direct human validation. However, we are limited by the availability of clinical samples since the intestinal tissues from patients with II/R-

related conditions (e.g., acute mesenteric ischemia, volvulus) are rarely available for research due to the urgency of clinical intervention and ethical constraints on tissue collection. In the future, we will strengthen our cooperation with clinical surgeons and actively validate basic medical achievements in clinical pathological samples.

References:

- 1 Meng, Y., Sandow, J. J., Czabotar, P. E. & Murphy, J. M. The regulation of necroptosis by post-translational modifications. *Cell Death Differ* **28**, 861–883, doi:10.1038/s41418-020-00722-7 (2021).
- 2 Zhou, Z. *et al.* Stress-induced epinephrine promotes epithelial-to-mesenchymal transition and stemness of CRC through the CEBPB/TRIM2/P53 axis. *J Transl Med* **20**, 262, doi:10.1186/s12967-022-03467-8 (2022).
- 3 Yin, H. *et al.* GPER promotes tamoxifen-resistance in ER+ breast cancer cells by reduced Bim proteins through MAPK/Erk-TRIM2 signaling axis. *Int J Oncol* **51**, 1191–1198, doi:10.3892/ijo.2017.4117 (2017).
- 4 Schonrock, N., Humphreys, D. T., Preiss, T. & Gotz, J. Target gene repression mediated by miRNAs miR-181c and miR-9 both of which are down-regulated by amyloid-beta. *J Mol Neurosci* **46**, 324–335, doi:10.1007/s12031-011-9587-2 (2012).
- 5 Chen, X. *et al.* MicroRNA-145 targets TRIM2 and exerts tumor-suppressing functions in epithelial ovarian cancer. *Gynecol Oncol* **139**, 513–519, doi:10.1016/j.ygyno.2015.10.008 (2015).
- 6 Xu, K. *et al.* MicroRNA-145-5p targeting of TRIM2 mediates the apoptosis of retinal ganglion cells via the PI3K/AKT signaling pathway in glaucoma. *J Gene Med* **23**, e3378, doi:10.1002/jgm.3378 (2021).
- 7 Zhang, Z., Fu, X., Gao, Y. & Nie, Z. LINC01535 Attenuates ccRCC Progression through Regulation of the miR-146b-5p/TRIM2 Axis and Inactivation of the PI3K/Akt Pathway. *J Oncol* **2022**, 2153337, doi:10.1155/2022/2153337 (2022).
- 8 Fang, C. *et al.* MicroRNA-181c Ameliorates Cognitive Impairment Induced by Chronic Cerebral Hypoperfusion in Rats. *Mol Neurobiol* **54**, 8370–8385, doi:10.1007/s12035-016-0268-6 (2017).
- 9 Wei, X. *et al.* Construction of circRNA-based ceRNA network to reveal the role of circRNAs in the progression and prognosis of metastatic clear cell renal cell carcinoma. *Aging (Albany NY)* **12**, 24184–24207, doi:10.18632/aging.104107 (2020).
- 10 Yi, H., Luo, D., Xiao, Y. & Jiang, D. Knockdown of long non-coding RNA DLEU2 suppresses idiopathic pulmonary fibrosis by regulating the microRNA-369-3p/TRIM2 axis. *Int J Mol Med* **47**, doi:10.3892/ijmm.2021.4913 (2021).
- 11 Chen, T. *et al.* CircORC2 promoted proliferation and inhibited the sensitivity of osteosarcoma cell lines to cisplatin by regulating the miR-485-3p/TRIM2

- axis. *J Cell Commun Signal* **18**, e12029, doi:10.1002/ccs3.12029 (2024).
- 12 Lv, R., Lu, F. & Xu, S. Hsa_circ_0001361 facilitates cell progression and glycolytic metabolism in neuroblastoma via interacting with mir-490-5p to induce TRIM2 upregulation. *Metab Brain Dis* **38**, 1621-1632, doi:10.1007/s11011-023-01197-4 (2023).
- 13 Liu, L. *et al.* Long non-coding RNA NR2F1-AS1 promoted neuroblastoma progression through miR-493-5p/TRIM2 axis. *Eur Rev Med Pharmacol Sci* **24**, 12748-12756, doi:10.26355/eurrev_202012_24174 (2020).
- 14 Ray, R. *et al.* BNIP3 heterodimerizes with Bcl-2/Bcl-X(L) and induces cell death independent of a Bcl-2 homology 3 (BH3) domain at both mitochondrial and nonmitochondrial sites. *J Biol Chem* **275**, 1439-1448, doi:10.1074/jbc.275.2.1439 (2000).
- 15 Kubli, D. A., Ycaza, J. E. & Gustafsson, A. B. Bnip3 mediates mitochondrial dysfunction and cell death through Bax and Bak. *Biochem J* **405**, 407-415, doi:10.1042/BJ20070319 (2007).
- 16 Yano, M. *et al.* Functional analysis of human mitochondrial receptor Tom20 for protein import into mitochondria. *J Biol Chem* **273**, 26844-26851, doi:10.1074/jbc.273.41.26844 (1998).
- 17 Hendgen-Cotta, U. B. *et al.* Cytosolic BNIP3 Dimer Interacts with Mitochondrial BAX Forming Heterodimers in the Mitochondrial Outer Membrane under Basal Conditions. *Int J Mol Sci* **18**, doi:10.3390/ijms18040687 (2017).

Response to reviewers' comments for manuscript

Thank you for your constructive feedback on our manuscript and for granting us the opportunity to submit a revised version. We are grateful to the reviewers for their insightful comments, which have helped improve the quality of our work. Below, we provide a point-by-point response to all reviewer comments. All changes in the revised manuscript are highlighted in red text for ease of identification.

Reviewer #1 (Remarks to the Author):

The authors have responded to the reviewers' comments, and I believe the manuscript in its current form is nearly acceptable. However, I would like to offer a few minor comments, which the authors are not required to address:

1. Although the authors extensively revised the manuscript in response to the reviewers' suggestions, the rebuttal letter lacks clear indications of where the revised or added sentences appear in the revised text. While the changes are highlighted in red, this alone is insufficient. Providing specific line numbers from the revised manuscript would make it easier for reviewers to locate and assess the changes.

Response: We sincerely apologize for this oversight. In this revised rebuttal letter, we have now added exact line numbers (based on the clean version of the revised manuscript) for every modification. All changes are also highlighted in red in the revised manuscript. We appreciate this suggestion to enhance clarity.

2. With regard to the evaluation of cell death, measuring only LDH activity in the culture supernatant is not appropriate, as it does not indicate what percentage of cells underwent membrane rupture. Instead, the authors should quantify LDH release as a percentage of the maximum release, typically obtained by treating cells with Triton X-100 to induce complete membrane disruption.

Response: We agree with the reviewer's critical point and have now fully addressed this concern. In our revised Materials and Methods section: Total LDH content was measured by lysing control cells with 0.2% Triton X-100 (as described in our original text). LDH release was calculated as follows: LDH Release (%) = [(LDH Activity in

$$\frac{\text{Conditioned Supernatant} \times (\text{Volume of Supernatant Collected})}{[\text{LDH Activity in Conditioned Supernatant} \times \text{Volume of Supernatant} + (\text{LDH Activity in Lysate Supernatant} \times \text{Volume of Lysate})]} \times 100\%$$

This method is now clearly stated in:

Methods Section: Lines 170–192.

We confirm that all LDH data presented in the revised manuscript now report % LDH release, normalized to total cellular LDH content (as shown in Figure 3A, Figure 4A, Figure 8A, Figure 8F, and Figure 8K).

Reviewer #2 (Remarks to the Author):

No further questions.

Response: We thank the reviewer for the positive assessment of our manuscript.

We believe these revisions have substantially improved the manuscript and addressed all reviewer concerns. Thank you again for your time and valuable input.

Response to reviewers' comments for manuscript

We sincerely appreciate the constructive feedback from the reviewers and the editorial team, which has significantly strengthened our work.

Reviewer #1 (Remarks to the Author):

*The authors have addressed the reviewer's comments appropriately, and the manuscript is now suitable for publication in *Communications Biology*.*

Response: Thank you for the positive decision on our manuscript.

We are pleased to confirm that we have addressed all previous reviewer comments thoroughly in the prior revision. As requested, we have edited the manuscript to comply fully with *Communications Biology*'s formatting guidelines, including structure, referencing style, and Figures/Table presentation. Maximize accessibility and impact by ensuring clarity in language and refining the abstract for broader readability.

We will submit the revised manuscript promptly and ensure it aligns seamlessly with the journal's requirements. Should any additional minor adjustments be needed, we are fully committed to implementing them swiftly.

Thank you again for your support. We look forward to contributing to *Communications Biology* and are honored to share our findings with the scientific community.